# PDE-Controller: LLMs for Autoformalization and Reasoning of PDEs

**Mauricio Soroco** [* 1]   **Jialin Song** [* 1]   **Mengzhou Xia** [2]   **Kye Emond** [3 4]   **Weiran Sun** [3]   **Wuyang Chen** [1]

## Abstract

While recent AI-for-math has made strides in *pure* mathematics, areas of *applied* mathematics, particularly PDEs, remain underexplored despite their significant real-world applications. We present **PDE-Controller**, a framework that enables large language models (LLMs) to control systems governed by partial differential equations (PDEs). Our approach enables LLMs to transform informal natural language instructions into formal specifications, and then execute reasoning and planning steps to improve the utility of PDE control. We build a **holistic** solution comprising *datasets (both human-written cases and 2 million synthetic samples), math-reasoning models, and novel evaluation metrics*, all of which require significant effort. Our PDE-Controller significantly outperforms prompting the latest open-source and GPT models in reasoning, autoformalization, and program synthesis, achieving up to a 62% improvement in utility gain for PDE control. By bridging the gap between language generation and PDE systems, we demonstrate the potential of LLMs in addressing complex scientific and engineering challenges. We release all data, model checkpoints, and code at `https://pde-controller.github.io/`.

## 1. Introduction

Recent advancements have significantly enhanced capabilities of Large Language Models (LLMs) (McKinzie et al., 2025; Huang et al., 2023). LLMs possess pretrained common knowledge and solves daily life tasks that require **commonsense reasoning** without domain-specific expertise. *However*, this reliance on generalized knowledge exposes

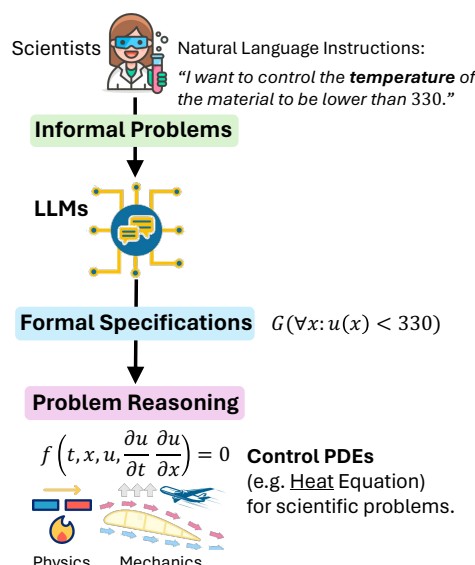

**Figure 1:** We build LLMs for automated, accelerated PDE control.

significant *weaknesses* in *complex domains*. LLMs struggle with precise mathematical reasoning (Mirzadeh et al., 2024; Feng et al., 2024; Ahn et al., 2024), understanding nuanced constraints (Williams & Huckle, 2024), or making decisions grounded in physical-world consequences (Wang et al., 2024; Jia et al., 2024; Cheng et al., 2024). Addressing these limitations will require enhancing LLMs with external tools or domain-specific reasoning.

Recent advancements in **AI-for-math** (Lu et al., 2022; Li et al., 2024) have significantly enhanced LLMs in logical, formal, and quantitative reasoning, particularly for pure mathematics (geometry, probability, calculus, algebra, number theory, and combinatorics). These efforts address challenges from grade school math (Cobbe et al., 2021) to the International Mathematical Olympiad (Hendrycks et al., 2021; Trinh et al., 2024). However, the advancement of LLMs for **applied mathematics**, such as partial differential equations (**PDEs**), remains underexplored. Unlike pure mathematics for abstract theory, *applied mathematics directly addresses practical challenges, bridging theory and real-world needs.* For example, PDEs are fundamental in modeling physical dynamics (aerospace engineering, quantum mechanics, fluid dynamics), providing a framework to understand and control systems. Integrating LLMs into applied mathematics, particularly for **PDE control**, holds substantial potential for advancing scientific and engineering applications.

---
[*]Equal contribution  [1]School of Computing Science, Simon Fraser University  [2]Department of Computer Science, Princeton University  [3]Department of Mathematics, Simon Fraser University  [4]Department of Physics, Simon Fraser University. Correspondence to: Wuyang Chen <wuyang@sfu.ca>.

*Proceedings of the 42nd International Conference on Machine Learning*, Vancouver, Canada. PMLR 267, 2025. Copyright 2025 by the author(s).

Solving PDE control problems has never been easy. Traditional approaches like optimization (McNamara et al., 2004) and formal methods (Alvarez, 2020) suffer from two bottlenecks: *First*, manual PDE control requires significant human efforts to understand problem descriptions and formalize into specifications (Alvarez, 2020). *Second*, PDE control requires highly specialized knowledge on both coding and physics, which are challenging even for seasoned mathematicians and engineers. The alternative is to leverage pretrained LLMs. However, commonsense logic captured by popular language datasets (Gao et al., 2020; Weber et al., 2024) largely deviates from scientific reasoning that requires math and physics backgrounds, leading to poor performance (Table 4 and 9). Moreover, unlike conventional programming languages (Python, Java, etc.) that are rich on GitHub (Chen et al., 2021), formal languages and special libraries required by PDE control (Fig. 4) are mainly used by relatively few mathematicians and engineers, resulting in limited language datasets. Thus, our core question is:

> ***Can LLMs control PDEs with scientific reasoning?***

In this work, we aim to **advance AI-for-math for PDEs**, making open-loop PDE control automated and accessible to broad scientific practitioners with reduced efforts (Fig. 1). Our framework, **PDE-Controller**, provides affirmative answers. PDE-Controller enables autoformalization of informal PDE control problems into formal specifications and executable code, and enhances scientific reasoning by proposing novel subgoals to improve open-loop control utility. This leads to a new methodology for integrating LLMs into scientific computing. Our technical contributions are summarized below:

1. **Scientific Reasoning.** Our PDE-Controller achieves up to a 62% improvement in PDE control utility gain, compared to prompting the latest LLMs. This demonstrates a promising approach to future LLM-based scientific reasoning. Our Controller is trained via reinforcement learning from human feedback (RLHF), with rewards derived from PDE simulations labeled as win or lose.

2. **Autoformalization and program synthesis.** We train LLMs via supervised fine-tuning (SFT) to automatically formalize PDE control problems, transforming informal natural language descriptions into formal specifications (over 64% accuracy) and executable programs that integrate with external tools (over 82% accuracy).

3. **New Datasets.** We build the *first* comprehensive datasets for PDE control designed for LLMs, including over *2 million* samples of natural and formal language, code programs, as well as PDE control annotations. We also collect manually written samples by *human volunteers* to evaluate LLMs in real-world scenarios. Our novel dataset will serve as a high-quality testbed for future research in AI for PDE reasoning.

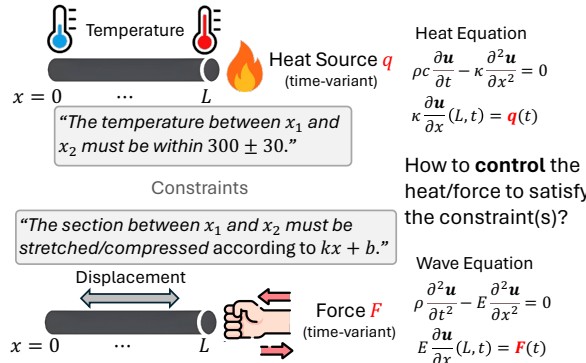

**Figure 2:** PDE control adjusts inputs (heat, force) to ensure systems (modeled by PDEs) satisfy spatiotemporal constraints.

## 2. Preliminaries

### 2.1. Background of PDE Control

Partial differential equations (PDEs) model nearly all of the physical systems and processes of interest to scientists and engineers. PDE control involves adjusting external inputs like heat or force to guide a system governed by physical laws (PDEs) to meet specific goals or constraints. For example, heat flow and mechanical stretching/compression in a rod are modeled and controlled by the heat and wave equations (Fig. 2). The goal is to maintain the rod's temperature or deformation within a safe range, which requires precise, time-varying control of the heat source and applied force. This is challenging because PDEs describe complex interactions across space and time, and small changes at one point can affect the entire system. Essentially, PDE control ensures the system behaves predictably and stays within desired limits. We provide more details in Appendix A.1.

### 2.2. Formal Methods for PDE Control

**Signal Temporal Logic.** Following previous works, we use signal temporal logic (STL) (Maler & Nickovic, 2004; Alvarez, 2020) to formally represent constraints in PDE control problems:

$$\phi = \mathcal{T}_{[t_1,t_2]} \left( \forall x \in [x_1, x_2], u(x) \lessgtr (ax + b) \right) \quad (1)$$

where $\mathcal{T} \in \{\mathbf{G}, \mathbf{F}\}$ and $\lessgtr$ indicates a choice from $\{<, >, =\}$. Specifically:

- Each STL $\phi$ defines a spatiotemporal constraint on the target variable $u$ (the quantity to be controlled, like temperature, displacement). For simplicity, we only consider time-invariant linear constraints $ax + b$. For example: $\forall x \in [x_1, x_2], \forall t \in [t_1, t_2], u(x, t) - \left(\frac{x}{2} + 300\right) \geq -3$.

- **G** ("globally") means the constraint holds *during* a specified interval. **F** ("eventually") means the constraint is satisfied *at least once* during the temporal interval.

- Composing multiple constraints can form more complicated constraints.

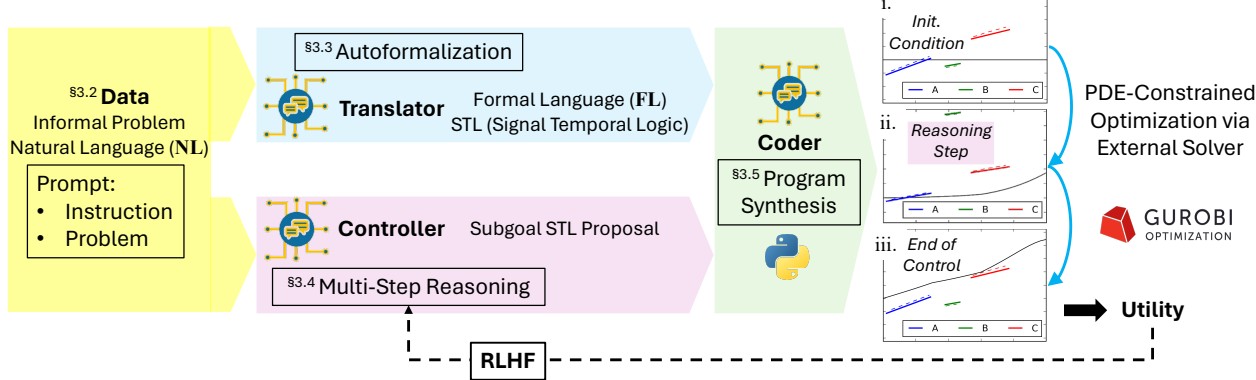

**Figure 3:** Overview of our PDE-Controller framework. The Translator directly autoformalizes an informal PDE control problem (yellow) into formal specifications with STL (blue). The Controller proposes novel STL subgoals (purple). Each STL is synthesized into specialized Python programs by the Coder (green) and optimized externally (white). From the initial condition (i.), our PDE reasoning optimizes a subgoal (ii.) before the original problem, improving the utility at the end of control (iii.). We train the Controller with reinforcement learning from human feedback (RLHF).

**Table 1:** Overview of our LLM components. "NL": natural language. "STL": Signal Temporal Logic.

| LLM | Purpose | Training Method | Input | Output | Evaluation Metric (s) |
|---|---|---|---|---|---|
| Translator (Sec. 3.3) | Autoformalize constraints from informal NL into formal STL | SFT | NL | STL | IoU: true vs. predicted constraints |
| Controller (Sec. 3.4) | Propose subgoal STLs as intermediate reasoning steps to improve the final control utility | RLHF (DPO) | NL + STL | Subgoal STL | Success Rate (P) Utility Gain ($\Delta r$) |
| Coder (Sec. 3.5) | Generate Python code for PDE solver (Gurobi) | SFT | NL + STL | Python code | Executability (%), Utility RMSE (on $r(\phi)$) |

Representing PDE constraints using STL can clearly and precisely express complex specifications into logical formulas and thus remove possible ambiguity in informal natural language. In addition to binary semantics ("satisfied" or "unsatisfied") defined above, STL admits continuous semantics as **utility** (Kress-Gazit et al., 2009; Donzé & Maler, 2010). The utility an STL $\phi$ achieves (via simulation and optimization) can be denoted as $r(\phi) \in \mathbb{R}$. Please refer to Appendix A.2 for the calculation of $r(\phi)$.

**Problem Example: Heat Equation.** Consider a metallic rod of 100 mm. The temperature at one end of the rod is fixed at 300K, a heat source is applied to the other end. The temperature of the rod follows a heat equation. We want the temperature of the rod to be within 3K of the linear constraint $\mu(x) = \frac{x}{4} + 300$ at all times between 4 and 5 seconds between 30 and 60 mm. Furthermore, the temperature should never exceed 345K at the point where the heat source is applied ($x = 100$). We can formulate this specification using the following composite STL formula:

$$\phi = \mathbf{G}_{[4,5]} \left( \left( \forall x \in [30, 60] : u(x) - \left( \frac{x}{4} + 303 \right) < 0 \right) \land \right.$$
$$\left. \left( \forall x \in [30, 60] : u(x) - \left( \frac{x}{4} + 297 \right) > 0 \right) \right) \land$$
$$\mathbf{G}_{[0,5]}(\forall x \in [100, 100] : u(x) - 345 < 0).$$

For more examples, please read Appendix B.

### 2.3. Optimization

To solve the PDE control problem defined by initial conditions and STL constraints (Eqn. 1), the PDE is first discretized into a set of difference equations with the finite element method. Then, together with STL constraints, they are formulated into a mixed integer linear program (MILP) (Sadraddini & Belta, 2015; Alvarez, 2020) where the utility $r$ can be optimized via external optimizers like Gurobi (Gurobi Optimization, LLC, 2024). After the optimization, if $r > 0$, that means the system successfully satisfies $\phi$ using the control input (i.e. constraints are successfully met). This is a non-convex optimization problem. Please see Appendix A.3 for detailed formulations.

## 3. Methods

Our core aim is to automate the formalization and reasoning of PDE control problems using large language models.

### 3.1. Overview

**Problem Definition.** As introduced in Sec. 2, the input of a PDE control problem is the natural language describing the initial state, system conditions, and target constraints. The solution is a time-variant trajectory of the control input. The final output we collect is the control utility $r(\phi) \in \mathbb{R}$.

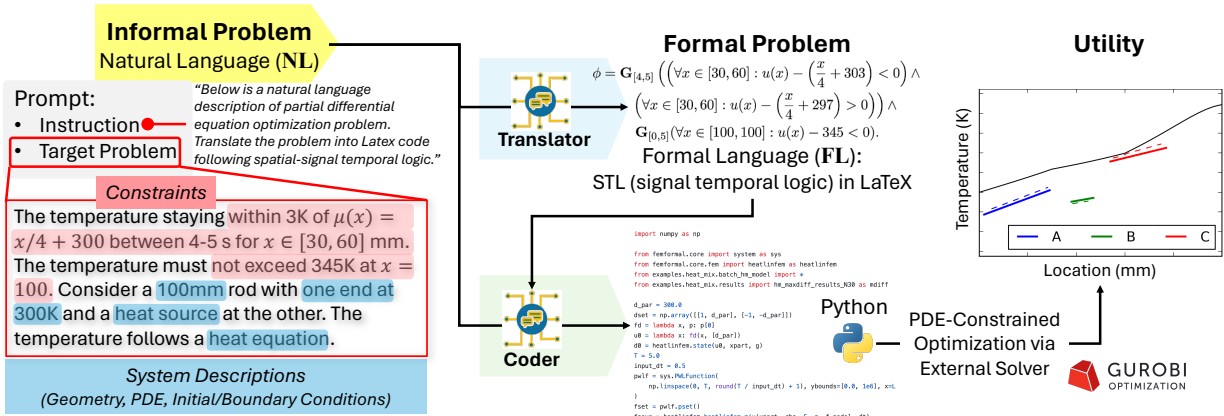

**Figure 4:** Workflow for supervised fine-tuning (SFT) of autoformalization (Translator LLM) and program synthesis (Coder LLM). Note that the utility is only used for evaluation and not used for SFT. Without reasoning, the Translator and Coder try to faithfully and directly solve the original problem.

**Framework Overview (Fig. 3).** We automate the PDE control problem with the following four steps:

1. **Input.** Our LLM prompt is composed of 1) an instruction (to prompt the LLM to formalize the problem, perform reasoning, or synthesize Python programs) and 2) the target PDE control problem. We build a large-scale dataset to support diverse inputs and prompts (Sec. 3.2).

2. **Autoformalization.** Our Translator LLM will extract information about the problem constraints from the prompt, and formalize these into STLs (Sec. 3.3).

3. **Reasoning.** Before directly solving the target PDE control problem, our Controller LLM will propose novel STLs as subgoals. The aim is to better solve the original PDE control problem by leveraging this subgoal as an intermediate step, i.e., we will first solve the problem defined by this subgoal which leads to a new system state, then solve the original problem (Sec. 3.4).

4. **Program Synthesis.** Our Coder LLM will take **both** the prompt and the formalized STLs as inputs, and generate Python code to be fed into the external tool (Gurobi optimizer (Gurobi Optimization, LLC, 2024)) to solve the PDE control problem (Sec. 3.5).

**3.2. Principled Data Synthesis with Augmentations**

Training LLMs for PDE problems requires diverse data, but existing math reasoning datasets (Cobbe et al., 2021; Yang et al., 2024; Glazer et al., 2024) lack large-scale corpora for PDEs. We collect key representative problems for PDE control, use them as templates, and augment them to ensure sufficient quantity and diversity.

**Overview.** As shown in Fig. 4 left, each PDE control problem consists of two components: 1) constraints (red background), 2) system descriptions (blue background, including PDE, geometry, initial/boundary conditions). We generate dataset in three steps:

**1) Constraints: Principled Syntax Formats via STL.** We first design and organize eligible syntax formats for generating STLs in two levels:

- For each constraint, the format in Eqn. 1 leads to 6 different syntax formats ($\{\mathbf{G}, \mathbf{F}\} \times \{<, >, =\}$).

- We consider STLs up to 3 constraints. For 2 constraints, we connect $\phi_1$ and $\phi_2$ with logical connectives $\wedge$ or $\vee$. For 3 constraints, we consider both logical connectives and operator precedence via parentheses: $\phi_1 \vee \phi_2 \vee \phi_3, \phi_1 \wedge \phi_2 \wedge \phi_3, (\phi_1 \vee \phi_2) \wedge \phi_3, \phi_1 \vee (\phi_2 \wedge \phi_3), (\phi_1 \wedge \phi_2) \vee \phi_3, \phi_1 \wedge (\phi_2 \vee \phi_3)$. In total, STLs with 1, 2, 3 constraints will respectively result in 6, 72, 1296 unique syntax formats, i.e. 1374 in total (LaTeX in Fig. 4 middle top).

At this moment, these STLs are still abstract, with hyperparameters like $a, b, x_1, x_2, t_1, t_2$ in Eqn. 1 not yet realized.

**2) System Descriptions: Sampling Initial Conditions and Hyperparameters.** To realize each STL format into a concrete problem, we need to fill numerical values for its initial conditions and hyperparameters (blue highlight in Fig. 4 left). We describe our sampling distributions in Appendix C.1 for the following aspects:

- Initial conditions: the initial temperature or displacement of the system at the beginning of the PDE simulation.

- Simulation domains: the spatial range $x_{\max}$ and temporal range $t_{\max}$.

- Physical properties: such as density, specific heat capacity, thermal conductivity.

- Coefficients: the linear parameters ($a$, $b$) and spatiotemporal ranges ($x_1, x_2, t_1, t_2$) for constraints.

**3) Dataset Synthesis with Augmentations.** We synthesize STLs into informal natural language descriptions equipped with system descriptions. For each problem, we

**Table 2:** Our dataset for autoformalization and program synthesis.

| Num. Constraints | 1 | 2 | 3 | Total Num. |
|---|---|---|---|---|
| STLs | 6 | 72 | 1296 | 1374 |
| Heat (Train) | 3840 | 45792 | 817776 | 867408 |
| Heat (Test) | 960 | 11448 | 204768 | 217176 |
| Wave (Train) | 3840 | 45504 | 795744 | 845088 |
| Wave (Test) | 960 | 11304 | 196992 | 209256 |

also synthesize ground-truth Python code for optimization with the Gurobi solver (Fig. 4 middle bottom). These synthesized samples give us a one-to-one mapping from the informal problem (natural language) to its STL and Python code. Details about our dataset synthesis, are in Appendix C.2. More importantly, to further promote the diversity of our informal natural language problems, we use ChatGPT (GPT4o-mini) for augmentation by rephrasing without affecting semantics. In particular, we prompt five paraphrases from ChatGPT for every synthesized informal description. Please see Appendix C.3 for an example of augmentation via ChatGPT.

As shown in Table 2, in total we have $n = 2.13$ million triplets of (natural language, STL, Python) samples. We merge the training set for both heat and wave problems for the training of Translator and Coder.

**Real-World Manually Written PDE Control Problems.** To evaluate our LLMs on real-world problems with high variance and noise, we collect PDE control samples designed by humans via questionnaires. To ensure data quality, we recruit undergraduate and graduate students as participants with highly relevant backgrounds (majoring in Math, Electic Engineering, Computer Science, Physics). During four questionnaire sessions (one hour each), we provide comprehensive introductions to the settings of our PDE control problems, with concrete examples and interactive communication. We collect 17 manually written heat problems and 17 wave problems. Details of this collection and differences between our training set are shown in Appendix D.

### 3.3. Autoformalization

After building our dataset, we train our Translator to extract constraints from informal natural language and autoformalize into STLs. Our LLM needs to (1) separate constraints from system descriptions, (2) align informal definitions and concepts to formal STL syntax, and (3) connect multiple constraints with correct logic operators. This task is further complicated by context changes in different PDEs.

We leverage a pretrained MathCoder2-DeepSeekMath-7B (Lu et al., 2024) checkpoint (MathCoder2), to fine-tune using LoRA (Hu et al., 2021) and supervised fine-tuning (SFT) with the cross-entropy loss. This measures the error in predicting each token in the output formal sequence ($\widehat{\mathbf{FL}}$)

given tokens in the informal input (**NL**). It is defined as:

$$\mathcal{L}_{SFT}^{\text{translator}} = -\sum_{i=1}^{n} \log P\left(\widehat{\mathbf{FL}}_i \mid \mathbf{NL}_i, \boldsymbol{\theta}_{\text{translator}}\right) \quad (2)$$

### 3.4. PDE Reasoning via Controller LLM

#### 3.4.1. WHAT IS REASONING FOR PDE CONTROL?

Beyond autoformalization, an important question is: can LLMs show **reasoning and planning** capabilities on scientific problems like PDE control, where pretrained commonsense knowledge may not be helpful?

**Problem Definition.** As described in Sec. 2.3, the PDE control problem is non-convex. Directly optimizing the target anchor problem may lead to suboptimal solutions or intractability due to potentially poor initial conditions, loss landscape barriers, local minima, etc. Inspired by recent works in robotics (Lin et al., 2024; Wang et al.) and AI-for-math (Zawalski et al., 2022; Zhao et al., 2023b;a) where subgoals are decomposed from the original problem, we propose the following **PDE control reasoning strategy**: *The solution to a PDE control problem can be improved by decomposing it into subgoals to be optimized sequentially.*

**PDE Control Reasoning.** We design the following reasoning steps for solving PDE control problems:

1. Given a target PDE control problem $\phi$ (dubbed "anchor"), we sample its STL constraints into a subgoal STL, $\phi'$, of different spatiotemporal constraints.

2. We directly solve the anchor problem ($\phi$) and collect its utility $r(\phi)$ and runtime cost $t$.

3. We solve $\phi' \rightarrow \phi$: We optimize $\phi'$, apply the system state as the new initial condition[1] solve the anchor problem $\phi$, and collect the final utility $r(\phi|\phi')$.

4. If $r(\phi|\phi') > r(\phi)$ (the utility of directly solving $\phi$), we call $\phi'$ a successful reasoning.

5. We repeat the above steps multiple times to calculate the expected performance gain.

Whether the subgoal reasoning step can be satisfied (i.e. whether or not $r(\phi') > 0$ or not) is not our concern. All we pursue is a new initial condition of the system that can better solve the original problem. As such, the time constraints of the subgoal should apply in the period prior to the anchor constraints and within the global time frame. Please see Fig. 6 for concrete examples where $r(\phi|\phi')$ outperforms $r(\phi)$, and more examples in Appendix B.

---

[1]To avoid long runtime of subgoal STLs, we set the Gurobi runtime threshold to 120 seconds for solving $\phi'$.

### 3.4.2. BUILDING THE CONTROLLER LLM.

Having defined PDE control reasoning in Sec. 3.4.1, our question is: *How to develop LLMs to perform PDE control reasoning and automatically decompose subgoals?*

In this section, we explain how to train a Controller LLM via reinforcement learning with human feedback (RLHF) that can perform PDE control reasoning. We illustrate our training strategy in Fig. 5.

**1) Preparing Preference Dataset.** We first build a dataset of paired STLs $(\phi'^{(w)}, \phi'^{(l)})$ as win-lose pairs, where $\phi'^{(w)}$ is preferred over $\phi'^{(l)}$. This dataset is used to train our Controller LLM via RLHF. To build this dataset, given an anchor (target) STL $\phi$, we randomly sample $\phi'$ based on $\phi$, solve both, and collect their utilities.

It is up to human preference to determine which STL is favored. In this work, we prefer $r(\phi|\phi'^{(w)}) > r(\phi) > \phi'^{(l)}$. In total, we collect 10772 pairs of win-lose STLs.

**2) Fine-tuning Controller LLM with RLHF.** During training, our Controller loads the Translator's pretrained checkpoint (inheriting the ability to faithfully formalize natural language into STLs). We train Controller with Direct Preference Optimization (DPO) (Rafailov et al., 2024). Unlike the Translator's direct formalization into true STLs, we prompt our Controller to make modifications in proposing new STL variants. We leverage win-lose pairs in our preference dataset as feedback to fine-tune the generative distribution of the Controller to shift to the preferred STL:

$$\mathcal{L}_{DPO}^{\text{controller}} = -\mathbb{E}_{(\mathbf{NL}, \phi'^{(w)}, \phi'^{(l)})} \left[ \log \sigma \left( \beta \log \frac{P\left(\phi'^{(w)} \mid \mathbf{NL}\right)}{P_{\text{ref}}\left(\phi'^{(w)} \mid \mathbf{NL}\right)} \right. \right.$$

$$\left. \left. - \beta \log \frac{P\left(\phi'^{(l)} \mid \mathbf{NL}\right)}{P_{\text{ref}}\left(\phi'^{(l)} \mid \mathbf{NL}\right)} \right) \right] + \lambda \log P\left(\widehat{\mathbf{FL}}|\mathbf{NL}\right).$$

$$(3)$$

$\sigma$ is the sigmoid function. We load the pretrained checkpoint of our Translator LLM as the frozen reference model ("ref"), which serves as the KL divergence target in DPO training. $P$ and $P_{\text{ref}}$ indicate the generative probability of Controller and Translator respectively ($\theta$ omitted for simplicity). $\beta$ controls the deviation of the Controller from the reference Translator. To further avoid the degradation of Controller's generation, we regularize it with the SFT loss, with a weight of $\lambda$ (Pang et al., 2024), which we also found prevented overfitting compared to the DPO loss alone.

### 3.5. Program Synthesis

Finally, taking both the prompt (instruction plus the target PDE control problem) and the formalized STL as inputs, we train a Coder LLM with supervised fine-tuning (SFT) to synthesize Python programs provided to the PDE simulator

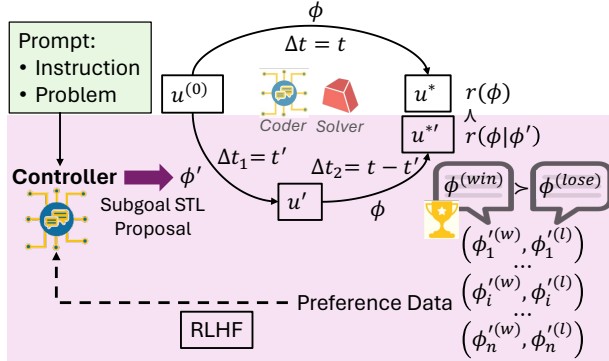

**Figure 5:** Learning PDE control reasoning via RLHF. Given the input prompt, our Controller LLM trained with preference data via reinforcement learning, will propose a subgoal STL $\phi'$. From the initial condition $u^{(0)}$, the PDE system is controlled by $\phi'$ to reach state $u'$, and then further controlled by the original STL $\phi$ to reach the final state $u^{*'}$. We expect the utility achieved via this reasoning, $r(\phi|\phi')$, to outperform $r(\phi)$ achieved by directly solving $\phi$.

and Gurobi optimizer to solve the PDE control problem.

$$\mathcal{L}_{SFT}^{\text{coder}} = -\sum_{i=1}^{n} \log P\left(\widehat{\mathbf{Code}}_i \mid \widehat{\mathbf{FL}}_i, \mathbf{NL}_i, \theta_{\text{coder}}\right) \quad (4)$$

Similar to Translator, we merge the training set for both heat and wave problems for the Coder's fine-tuning.

## 4. Experiments

We study 1D heat and wave problems as pioneering showcases. All our models are fine-tuned from MathCoder2, and we compare against few-shot evaluations of MathCoder2, GPT 4o, and GPT o1-mini (Achiam et al., 2023). Please read Appendix E for model and training details.

### 4.1. Accurate Autoformalization and Program Synthesis

We first evaluate the performance of our Translator for autoformalization and Coder for program synthesis.

**Evaluation Metrics.** To progressively evaluate the performance of our Translator and Coder LLMs in a *decoupled and fine-grained* manner, we propose to leverage multiple metrics for different purposes, as summarized in Table 3. Among all metrics, the "Utility RMSE" (lower the better) is the most important final performance. However, it is important to emphasize that we can only calculate RMSE for executable Python programs. Consequently, if we observe a coder achieving a low RMSE but also low executability, it still implies poor quality.

---

[2]For example, switching from $\geq$ to $\leq$ will lead to a completely different constraint, but the STL itself is still a valid logic.

[3]Remarks for Table 4 and 5: 1) Since IoU $\in [0, 1]$, we set the IoU of any invalid STL as 0. 2) To isolate the evaluation of program synthesis without being distracted by possibly generated bad STLs, when calculating the executability and utility RMSE,

**Table 3:** Metrics for the evaluation of autoformalization (Translator) and program synthesis (Coder).

| Purpose | Metric |
|---|---|
| LLM-generated STL ($\hat{\phi}$) could be valid, but there might be semantic mistakes[2]. | **IoU**: Intersection over union (satisfying areas) between the true STL, $\phi$, and the LLM's generation, $\hat{\phi}$. |
| Python programs generated by Coder may not be runnable due to bugs. | **Executability**: The ratio of executable programs to the total. *This doesn't ensure the executed result (utility $r$) is correct.* |
| Compare the final PDE control utility $r(\hat{\phi})$ to true utility $r(\phi)$. | **Utility RMSE**: Relative mean square error on utility. |

**Results[3].** The autoformalization can be evaluated using the intersection over union (IoU) between the predicted and target STLs (constraints). The code generation should aim for high executability and low utility RMSE simultaneously[4]. As in Table 4, our Translator and Coder achieve the best across all metrics with low deviations, indicating strong and reliable autoformalization and program synthesis.

We further evaluate manually written problems. As shown in Table 5, the autoformalization may produce STLs with worse quality (lower IoU), suffering from noisy and unstructured texts written by humans. Our Translator and Coder generally outperform other baselines, with the accuracy of autoformalization over 64% (STL's IoU) and program synthesis over 82% (code executability). Importantly, despite the low utility RMSE of GPT o1-mini, since it suffers from poor executability (only 39.22%), its RMSE cannot faithfully characterize its stability in the real world.

**Table 4:** Autoformalization and program synthesis. Deviations over 3 seeds are in parentheses. **Bold** indicates the best, underline denotes the runner-up.

| PDE | Model | IoU (↑) (Translator) | Executability (↑) (Coder) | **Utility RMSE** (↓) |
|---|---|---|---|---|
| Heat | Ours | **0.992** (0.07) | **0.9978** (0.0015) | **0.0173** (0.0065) |
| | MathCoder2 | 0.772 (0.35) | 0.9592 (0.0166) | 0.2058 (0.0672) |
| | GPT (4o) | - | 0.5807 (0.3244) | 0.0445 (0.0437) |
| | GPT (o1-mini) | - | 0.3561 (0.2857) | 0.0898 (0.0165) |
| Wave | Ours | **0.992** (0.07) | **0.9620** (0.0098) | **0.0076** (0.0011) |
| | MathCoder2 | 0.772 (0.35) | 0.9340 (0.0242) | 0.1089 (0.0485) |
| | GPT (4o) | - | 0.6799 (0.2523) | 0.0868 (0.0500) |
| | GPT (o1-mini) | - | 0.4041 (0.2771) | 0.0757 (0.0149) |

LLMs are provided with true STLs in their prompts, instead of LLM-generated STLs. Table 15 and 16 show end-to-end results but cannot decouple the quality of program synthesis. 4) The utility RMSE is only calculated for executable Python programs.

[4]Otherwise, the code generation might trivially generate for example a runnable "`return 0`", which is obviously incorrect; or it might get lucky and generate the correct code for only a very limited set of problems.

**Table 5:** Autoformalization and program synthesis on manually written data (Sec. 3.2). Deviations over 3 seeds are in parentheses. **Bold** indicates the best, underline denotes the runner-up.

| PDE | Model | IoU (↑) (Translator) | Executability (↑) (Coder) | **Utility RMSE** (↓) |
|---|---|---|---|---|
| Heat | Ours | **0.7108** (0.0043) | 0.8235 (0.0) | 2.4687 (0.0) |
| | MathCoder2 | 0.3383 (0.068) | **0.9804** (0.0277) | **0.0004** (0.0005) |
| | GPT (4o) | - | 0.4314 (0.3328) | 1.8555 (0.0136) |
| | GPT (o1-mini) | - | 0.3530 (0.3050) | 0.1738 (0.0207) |
| Wave | Ours | **0.6493** (0.0) | **1.0** (0.0) | 0.0119 (0.0) |
| | MathCoder2 | 0.1953 (0.045) | **1.0** (0.0) | 0.0129 (0.0012) |
| | GPT (4o) | - | 0.5882 (0.4437) | 0.0105 (0.0) |
| | GPT (o1-mini) | - | 0.3922 (0.4160) | **0.0098** (0.0) |

**Generalization to Unseen STL Formats.** Although we only collect data with no more than three STLs, it is still possible to scale up our method to more constraints. To evaluate our generalization to unseen STL formats, we extend our method to unseen 4-constraint STLs. We evaluate our Translator (IoU) and Coder (executability & utility) over 5 problems each for heat and wave problems in Table 6, and compare against MathCoder2. We can see that our method generalizes much better than MathCoder2. Neither model has seen problems with 4 STL constraints. MathCoder2 is evaluated with 2 in-context examples of 4 constraints.

**Table 6:** Test different models on unseen 4-constraint STL formats, based on IoU, Executability, and Utility RMSE. Bold indicates the best. Deviations over 5 seeds are in parentheses.

| PDE | Model | IoU (↑) (Translator) | Executability (↑) (Coder) | **Utility RMSE** (↓) |
|---|---|---|---|---|
| Heat | Ours | **0.934** (0.0) | **0.8** (0.0) | **0.0** (0.0) |
| | MathCoder2 | 0.8154 (0.0) | 0.6 (0.0) | 0.2600 (0.0) |
| Wave | Ours | **1.0** (0.0) | **0.8** (0.0) | **0.1515** (0.0) |
| | MathCoder2 | 0.9690 (0.0) | **0.8** (0.0) | 0.2393 (0.2268) |

**Ablation Comparisons.** We further demonstrate the necessity of our Translator and Coder in Table 7. We observe the following by making comparisons:

- Our Translator has better autoformalization abilities, improving +28.5% IoU over MathCoder2.

- Our Coder is robust to noisy autoformalization:

  - Given ground truth STLs, our Coder has +4% better executability rate in generated Python than MathCoder2, and utility RMSE is 91.6% lower (better) than MathCoder2.
  - When switching from ground truth STL inputs to noisy Translator predictions, our Coder's utility RMSE is only 0.57% worse, indicating that our Coder is robust under noisy predicted STL inputs.

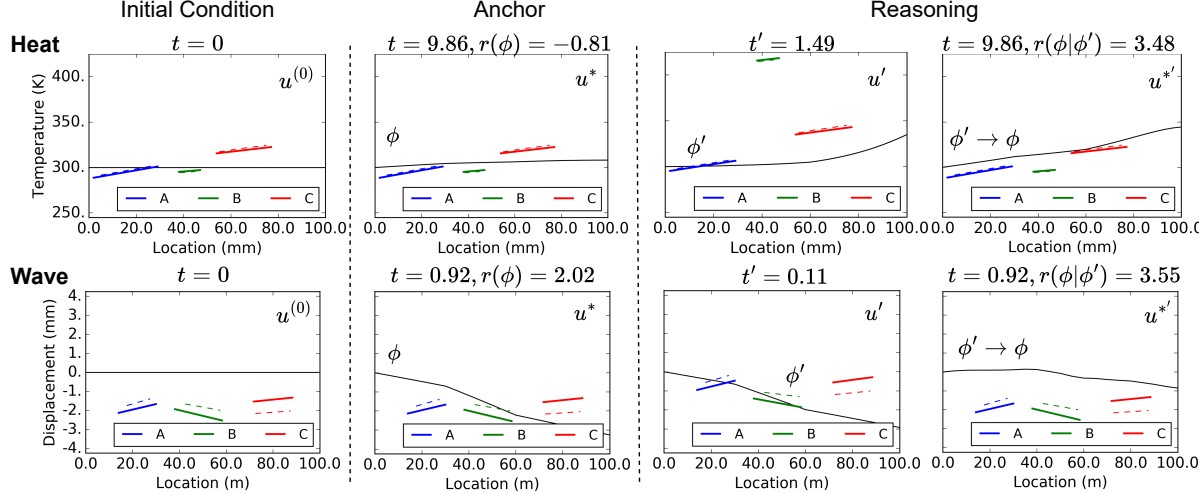

**Figure 6:** Case study of LLM reasoning for PDE control on heat (top) and wave (bottom) problems (symbols are aligned with Fig. 5). From left to right: Directly solving $\phi$ from the initial condition $u^{(0)}$ (1st column) yields $r(\phi)$ (2nd column); Reasoning: solving $\phi'$ from $u^{(0)}$ to get $u'$ (3rd column) then solving $\phi$ from $u'$ to get $r(\phi|\phi')$ (4th column). Black curves indicate the system's states (temperature for heat, displacement for wave) at $t$ or $t'$. Colored segments (A, B, C) are constraints, with dashes for inequalities ($\geq$ when dashes are above the solid, and vice versa). Although we plot constraints, *they may constrain temporal ranges* $[t_1, t_2] \not\ni t, t'$. Constraint STLs can be found in Appendix B.3.

**Table 7:** Ablation comparisons of translators and coders. Bold indicates the performance change relative to the row above.

| Method | Metric | Performance |
|---|---|---|
| Mathcoder's Translator (Table 4) Our Translator (Table 4) | IoU (↑) | 0.772 0.992 **(+28.5%)** |
| Mathcoder's Coder (Table 4) Our Coder (Table 4) | Executability (↑) | 0.9592 0.9978 **(+4.02%)** |
| Mathcoder's Coder (Table 4) Our Coder (Table 4) | Utility RMSE (↓) | 0.2058 0.0173 **(-91.6%)** |
| Translator STL → Coder (Table 15) Ground Truth STL → Coder(Table 4) | Utility RMSE (↓) | 0.0174 0.0173 **(-0.57%)** |

**Table 8:** Overview of our reasoning data. We threshold 3 difficulty levels of questions by the Success Rate $\overline{P}$ of random sampling.

| Heat | Training | Testing | Total |
|---|---|---|---|
| Num. $(\phi'^{(w)}, \phi'^{(l)})$ Pairs | 4813 | 1181 | 5994 |
| Easy $\overline{P} \in (0.8, 1)$ | 27.1% | 26.1% | 26.9% |
| Medium $\overline{P} \in (0.5, 0.8]$ | 37.3% | 37.8% | 37.4% |
| Hard $\overline{P} \in [0, 0.5]$ | 35.6% | 36.2% | 35.7% |

| Wave | Training | Testing | Total |
|---|---|---|---|
| Num. $(\phi'^{(w)}, \phi'^{(l)})$ Pairs | 3812 | 966 | 4778 |
| Easy $\overline{P} \in (0.88, 1)$ | 32.5% | 33.6% | 32.7% |
| Medium $\overline{P} \in (0.55, 0.88]$ | 33.1% | 32.5% | 33.0% |
| Hard $\overline{P} \in [0, 0.55]$ | 34.4% | 33.9% | 34.3% |

### 4.2. Improved Utility via PDE Reasoning of Controller

Beyond autoformalization and program synthesis, our most important contribution is the scientific reasoning on PDE problems by the Controller. In addition to the MathCoder2 and GPT models, we consider another baseline, *random sampling*, which naively generates reasoning steps by randomly sampling the anchor's constraints.

**Evaluation Metrics.** During inference, we sample from the Controller multiple times. We evaluate the reasoning performance on PDE control problems with two metrics:

- Success Rate $\overline{P}$: The percentage of sampled reasoning step ($\phi'$) that can improve the anchor problem ($\phi$), averaged across all anchor problems. $\overline{P} \triangleq \mathbb{E}_\phi P(\phi) = \mathbb{E}_\phi p\left(r(\phi|\phi') > r(\phi)\right)$.

- Utility Gain $\overline{\Delta r}$: The expected improvement in utility via a sampled reasoning step, averaged across all anchor

problems. $\overline{\Delta r} \triangleq \mathbb{E}_\phi \mathbb{E}_{\phi'} [r(\phi|\phi') - r(\phi)]$.

**Difficulty Levels.** Intuitively, some anchor problems are easy to improve via reasoning, while others may be more challenging to improve. To comprehensively study the performance of our controller, we design three difficulty levels based on the Success Rate, $\overline{P}$, of random sampling. We group problems by choosing thresholds on $\overline{P}$ of random sampling such that all difficulty levels share a balanced number of problems during testing. We overview our reasoning data in Table 8 and provide examples in Appendix B.

---

[5]In Table 9, to isolate the evaluation of the PDE reasoning quality ($\phi'$) from influence by autoformalization and code generation, we provide the true Python code for any valid generated subgoal STL $\phi'$. Table 17 shows end-to-end results but cannot decouple the quality of LLM PDE reasoning.

**Table 9:** Scientific reasoning over PDE control problems via our Controller LLM. Deviations over 5 seeds are in parentheses. "Valid STL $\phi'$ (%)": ratio of valid proposed subgoal STL $\phi'$ (i.e. without any syntax errors or improper time constraints). **Bold** indicates the best, underline denotes the runner-up. "x" indicates no valid STLs were generated for evaluation. "-" indicates not applicable.

| PDE | Difficulty Level | Success Rate $\overline{P}$ ($\uparrow$) | | | | | Utility Gain $\overline{\Delta r}$ ($\uparrow$) | | | | |
|---|---|---|---|---|---|---|---|---|---|---|---|
| | | Ours | Random Sampling | Math-Coderv2 | GPT (o1-mini) | GPT (4o) | Ours | Random Sampling | Math-Coderv2 | GPT (o1-mini) | GPT (4o) |
| Heat | Easy | **0.966** (0.0453) | 0.886 (0.0365) | 0.396 (0.2419) | x | 0.718 (0.1012) | **2.233** (0.6662) | 1.594 (1.0333) | 0.795 (07985) | x | 1.614 (0.0262) |
| | Medium | **0.877** (0.1010) | 0.694 (0.0909) | 0.340 (0.2324) | 0 (0) | 0.468 (0.0912) | **1.090** (0.5030) | 0.526 (0.6986) | -0.109 (0.2659) | -0.601 (0) | 0.222 (0.0560) |
| | Hard | **0.592** (0.1692) | 0.356 (0.1319) | 0.236 (0.1887) | 0 (0) | 0.469 (0.1147) | **1.035** (0.8486) | -0.380 (1.2688) | -0.964 (0.5101) | -1.489 (0) | 0.855 (0.0611) |
| | All | **0.812** (0.1052) | 0.645 (0.0864) | 0.324 (0.221) | x | 0.552 (0.1024) | **1.453** (0.6726) | 0.580 (1.0002) | -0.093 (0.5249) | x | 0.897 (0.0477) |
| Wave | Easy | 0.936 (0.0600) | 0.928 (0.0232) | 0.954 (0.0725) | x | **1** (0) | 1.423 (0.4135) | 1.110 (0.4986) | 1.601 (0.1794) | x | **1.706** (0) |
| | Medium | 0.833 (0.1009) | 0.737 (0.0894) | 0.769 (0.0362) | x | **0.933** (0.0408) | **0.901** (0.4389) | 0.704 (0.7072) | 0.830 (0.2122) | x | 0.652 (0.0191) |
| | Hard | 0.328 (0.1220) | 0.294 (0.1620) | 0.324 (0.1036) | x | **0.386** (0.1013) | **-0.349** (0.4357) | -0.531 (0.8028) | -0.670 (0.1945) | x | -0.609 (0.0196) |
| | All | 0.699 (0.0943) | 0.653 (0.0915) | 0.682 (0.0708) | x | **0.773** (0.0474) | **0.658** (0.4293) | 0.427 (0.6695) | 0.587 (0.1954) | x | 0.583 (0.0129) |
| Valid STL $\phi'$ (%) ($\uparrow$) | | **82.70** (1.97) | - | 42.45 (10.54) | 0.04 (0.10) | 2.55 (0.65) | **82.70** (1.97) | - | 42.45 (10.54) | 0.04 (0.10) | 2.55 (0.65) |

**Results**[5]. We observe the following from Table 9:

- In general, our Controller consistently outperforms other models for both heat and wave problems. While second to GPT-4o by wave success rates, GPT-4o suggests very few valid STLs caused by syntax errors or time constraints that occur after the anchor problem.

- In general, our Controller most significantly improves the utility. For example, on heat problems, our $\overline{\Delta r} = 1.453$, which improves 62% over the second-best (GPT 4o, $\overline{\Delta r} = 0.897$). Although no model improves utility for "hard"-level wave problems, our Controller still proposes the highest quality subgoals.

- Our Controller suggests far and away the highest ratio of valid STLs ($\phi'$), almost doubling the second-ranked MathCoder2.

- In most heat cases, MathCoder2 and GPT models are worse than random sampling. GPT o1-mini fails entirely due to invalid hallucinations in subgoal STL proposals.

On manual data, due to invalid subgoal STL proposals, all models fail to generate meaningful reasoning steps. Firstly, inconsistencies in the natural language of the manual data cause the models to generate invalid constraint values, such as different units to describe time within a sentence. Secondly, new notation such as ":=" in NL, result in unbalanced brackets, hallucinated numbers, and quantifiers. Thirdly, the models propose invalid time constraints for reasoning steps, which should occur before the anchor's time constraints.

**Case Study.** To better illustrate our PDE controller, we show one case for heat and wave in Fig. 6. In both examples, first optimizing the reasoning steps leads to new initial conditions that better solve each anchor problem.

## 5. Related Works

First, our paper belongs to the broader community of **auto-formalization in AI-for-math**. Autoformalization converts informal math into formal logic. LLMs like GPT-3.5/4o have been used to translate problems into Isabelle/Lean4 sketches (Zhou et al., 2024; Jiang et al., 2023). Verna-Copter (van de Laar et al., 2024) autoformalizes natural languages into STL with correctness checks. Hybrid methods combine manual and automated steps (Xin et al., 2024; Xiong et al., 2023; Murphy et al., 2024; Mishra et al., 2022). We introduce LLMs trained to autoformalize informal PDE control into STL. Second, our paper targets **PDE Controls**. PDEs model physical systems; control aims to ensure stability or optimize behavior (Alvarez, 2020; Wei et al.). Classical methods use adjoints (Lions, 1971), while learning-based ones use differentiable physics (Holl et al., 2020) or RL (Rabault et al., 2019). LLMs help with multimodal PDE surrogate modeling (Lorsung & Farimani, 2025). We reformulate PDE control into MILP via FEM (Sadraddini & Belta, 2015) and use LLMs to propose initializations for open-loop control, guided by a utility score (Appendix A.2). Finally, our work also studies **LLM-based Task Planning**. LLMs translate natural language to formal plans (e.g., LTL)(Pan et al., 2023). Recent work improves plan reliability (Wang et al.; Ren et al., 2023), with CLMASP refining outputs via ASP (Lin et al., 2024). We extend LLM-based reasoning to PDE control for the first time.

Please read Appendix G for more related works.

## 6. Conclusion

In this paper, we aim to transform the automation of PDE control problems. By utilizing LLMs to interpret natural language problem descriptions, formalize mathematical expressions, and apply scientific reasoning, we demonstrate that LLMs can fully automate PDE control while even improving control performance. Our research emphasizes the importance of LLMs in applied mathematics, and seeks to enable more accessible, scalable, and robust PDE solutions, ultimately expanding the practical reach and reliability of PDE applications across scientific and engineering domains.

## Acknowledgements

We thank Dr. Danqi Chen for her helpful comments. We thank all participants in our questionnaire for manually writing PDE control problems. For privacy reasons, we do not disclose their names.

## Impact Statement

This paper presents work whose goal is to advance the field of Machine Learning. There are many potential societal consequences of our work, none of which we feel must be specifically highlighted here.

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

# A. More Background on Formal Methods for PDE Control

## A.1. PDEs

We consider controlling systems governed by two popular PDEs (in 1D space):

- Heat Equation: Describes how heat diffuses through a material over time. Applications: temperature in buildings, pollution in the environment.

$$\rho c \frac{\partial \boldsymbol{u}}{\partial t} - \kappa \frac{\partial^2 \boldsymbol{u}}{\partial x^2} = 0,$$
$$\kappa \frac{\partial \boldsymbol{u}}{\partial x}(L, t) = \boldsymbol{q}(t),$$
$$\boldsymbol{u}(0, t) = g_0, \qquad \forall t \in [0, t_{\max}],$$
$$\boldsymbol{u}(x, 0) = u_0(x), \qquad \forall x \in [0, L]. \tag{5}$$

$\rho, c, \kappa > 0$: density, specific thermal capacity, thermal conductivity of the material respectively. $\boldsymbol{u}$ the spatiotemporal temperature of the material. $x \in [0, L]$ is the spatial location. $t \in [0, t_{\max}]$ is the time. $\boldsymbol{q}$ is the time-variant external heat source, applied at $x = L$. $g_0$ is the boundary condition applied at $x = 0$. $u_0$ is the initial condition (temperature).

- Wave Equation: Models the propagation of waves (sound, electromagnetic, or water waves). Applications: Acoustic control, vibration control in structures (e.g., bridges, buildings).

$$\rho \frac{\partial^2 \boldsymbol{u}}{\partial t^2} - E \frac{\partial^2 \boldsymbol{u}}{\partial x^2} = 0,$$
$$E \frac{\partial \boldsymbol{u}}{\partial x}(L, t) = \boldsymbol{F}(t),$$
$$\boldsymbol{u}(0, t) = g_0, \qquad \forall t \in [0, t_{\max}],$$
$$\boldsymbol{u}(x, 0) = u_0(x), \qquad \forall x \in [0, L]. \tag{6}$$

$\rho, E > 0$: density, Young's Modulus of the material respectively. $\boldsymbol{u}$ the spatiotemporal displacement of the material. $x \in [0, L]$ is the spatial location. $t$ is the time. $\boldsymbol{F}$ is the time-variant external fource, applied at $x = L$. $g_0$ is the boundary condition applied at $x = 0$. $u_0$ is the initial condition (displacement), and typically we set it as 0.

## A.2. Utility of STL

The continuous utility value of STL $r(\phi)$ is calculated with the following cases and rules:

$$r\left(\boldsymbol{u}, \boldsymbol{u} \geq ax + b, t\right) = \boldsymbol{u}(x, t) - (ax + b) \tag{7}$$
$$r\left(\boldsymbol{u}, \boldsymbol{u} \leq ax + b, t\right) = (ax + b) - \boldsymbol{u}(x, t) \tag{8}$$
$$r\left(\boldsymbol{u}, \phi_1 \wedge \phi_2, t\right) = \min\left\{r\left(\boldsymbol{u}, \phi_1, t\right), r\left(\boldsymbol{u}, \phi_2, t\right)\right\} \tag{9}$$
$$r\left(\boldsymbol{u}, \phi_1 \vee \phi_2, t\right) = \max\left\{r\left(\boldsymbol{u}, \phi_1, t\right), r\left(\boldsymbol{u}, \phi_2, t\right)\right\} \tag{10}$$
$$r\left(\boldsymbol{u}, \mathbf{F}_{[a,b)}\phi, t\right) = \sup_{t_f \in [t+a, t+b)}\left\{r\left(\boldsymbol{u}, \phi, t_f\right)\right\} \tag{11}$$
$$r\left(\boldsymbol{u}, \mathbf{G}_{[a,b)}\phi, t\right) = \inf_{t_g \in [t+a, t+b)}\left\{r\left(\boldsymbol{u}, \phi, t_g\right)\right\} \tag{12}$$

Here, Eq. 7 and 8 indicates the linear constraint we consider in Eq. 1.

## A.3. MILP Formulation of Control Synthesis

Solving the PDE control problem can be relaxed and formulated into a PDE-constrained optimization problem, which can be further solved by mixed-integer linear programming (MILP). We brief the high-level steps, and we recommend readers to read (Sadraddini & Belta, 2015; Alvarez, 2020) for more details.

We start with spatially and temporally discretizing the PDE. This is achieved by the finite element method (FEM):

1. The PDE is converted into its weak (variational) form by integrating against suitable test functions $\boldsymbol{v}(x)$. The purpose is to reduce the second-order derivatives of $\boldsymbol{u}$ to first derivatives so that we can obtain (linear) approximations to $\boldsymbol{u}$. This also simplifies boundary condition handling and smoothness requirements in the original PDE.

2. The spatial domain of the PDE's weak form is discretized by dividing it into small, simple geometric elements (intervals in 1D, triangles/quadrilaterals in 2D; tetrahedra in 3D), essentially forming a mesh, and choosing local basis functions on each element.

3. Contributions from local (spatial) elements are assembled across the entire mesh into a global linear system, often written as $M\dot{\tilde{u}} + K\tilde{u} = F$. The stiffness matrix ($K$) and mass matrix ($M$) encode the PDE's structure. The banded stiffness matrix ($K$) arises from terms involving derivatives (e.g., $\nabla u \cdot \nabla v$ in the weak form). The diagonal mass matrix ($M$) comes from non-derivative terms (e.g., $u \cdot v$). $F$ is the "force" vector (from external source terms/boundary conditions like heat and force). This step transforms the PDE into an ordinary differential equation (ODE) in time for the discretized spatial domain.

4. The temporal domain is further discretized using finite difference schemes, obtaining a set of difference equations that must be solved at each time step. This final step produces the final linear (or nonlinear) system of equations that is solved numerically to approximate the solution of the original PDE.

$$M\frac{\tilde{u}^{n+1} - \tilde{u}^n}{\Delta t} + K\tilde{u}^{n+1} = F^{n+1}. \tag{13}$$

This can be further rearranged to a linear system:

$$(M + \Delta t K)\tilde{u}^{k+1} = M\tilde{u}^k + \Delta t F^{k+1}. \tag{14}$$

At this moment, we can re-formulate the original PDE control problem into the following PDE-constrained optimization problem:

$$\max r\left(\phi, \tilde{u}\right) \tag{15}$$
$$\text{s.t. } (M + \Delta t K)\tilde{u}^{k+1} = M\tilde{u}^k + \Delta t F^{k+1}, \tag{16}$$
$$\tilde{u}^0 = \tilde{u}(0).$$

This formulation is equivalent to an MILP problem because:

- $\phi$ is only applied to limited spatiotemporal ranges. After the discretization of PDE, essentially $\phi$ is only selectively applied to certain areas of grids over our (1D) mesh. That means, we need **binary** variables to encode the absence/presence of $\phi$ over the spatiotemporal domain.

- The discretized PDE constraints Eq. 16 is linear in $\tilde{u}$. Additionally, based on Appendix A.2, the objective function $r(\phi, \tilde{u})$ (Eq. 15) is also linear in $\phi$ and $\tilde{u}$.

This MILP problem is non-convex, due to the $\min, \max$ operation and non-differentiability (Appendix A.2) of the objective function $r\left(\phi, \tilde{u}\right)$ in Eq. 15. This MILP problem is solved using the off-the-shelf Gurobi solver (Gurobi Optimization, LLC, 2024).

### A.4. Metrics - Additional Design Considerations

The careful design of our utility score, in Appendix A.2, can faithfully quantify whether the STLs (constraints) are fully met by the solution simulated by the Gurobi solver. This utility score is inherited from prior work (Alvarez, 2020).

Compared with standard metrics such as pass@k, our metrics can better quantify our LLMs' performance. IoU is designed to provide fine-grained quantification of the logic structures (junctions, temporal operators, boundary constraints) while pass@k lacking nuance will fail or succeed from trivial text-level differences in STL. We further explain the connection between our metrics and pass@k below:

- For autoformalization (Translator), IoU is equivalent to "average@k", it averages the alignment between predictions and targets over multiple generations and problems. We can discretize IoU into "pass@k" by considering only pass/fail cases based on token-level differences; but this ignores fine-grained quantifications of autoformalization. Namely, in tables 10 and 11 below, IoU and pass@k are not always aligned.

- For code generation (Coder), the executability metric is essentially pass@k from the executability perspective.

**Table 10:** Performance comparison for autoformalization (Translator) on Heat. Deviations over 3 seeds are in parentheses.

| Model | IoU | Pass@1 | Pass@2 | Pass@3 |
|---|---|---|---|---|
| Ours | 0.992 (0.07) | 0.978 (0.142) | 0.980 (0.134) | 0.982 (0.131) |
| MathCoder2 | 0.772 (0.35) | 0.538 (0.480) | 0.565 (0.484) | 0.583 (0.493) |

**Table 11:** Performance comparison for autoformalization (Translator) on Wave. Deviations over 3 seeds are in parentheses.

| Model | IoU | Pass@1 | Pass@2 | Pass@3 |
|---|---|---|---|---|
| Ours | 0.992 (0.07) | 0.971 (0.161) | 0.975 (0.152) | 0.977 (0.149) |
| MathCoder2 | 0.772 (0.35) | 0.3305 (0.4396) | 0.3726 (0.4663) | 0.3971 (0.4893) |

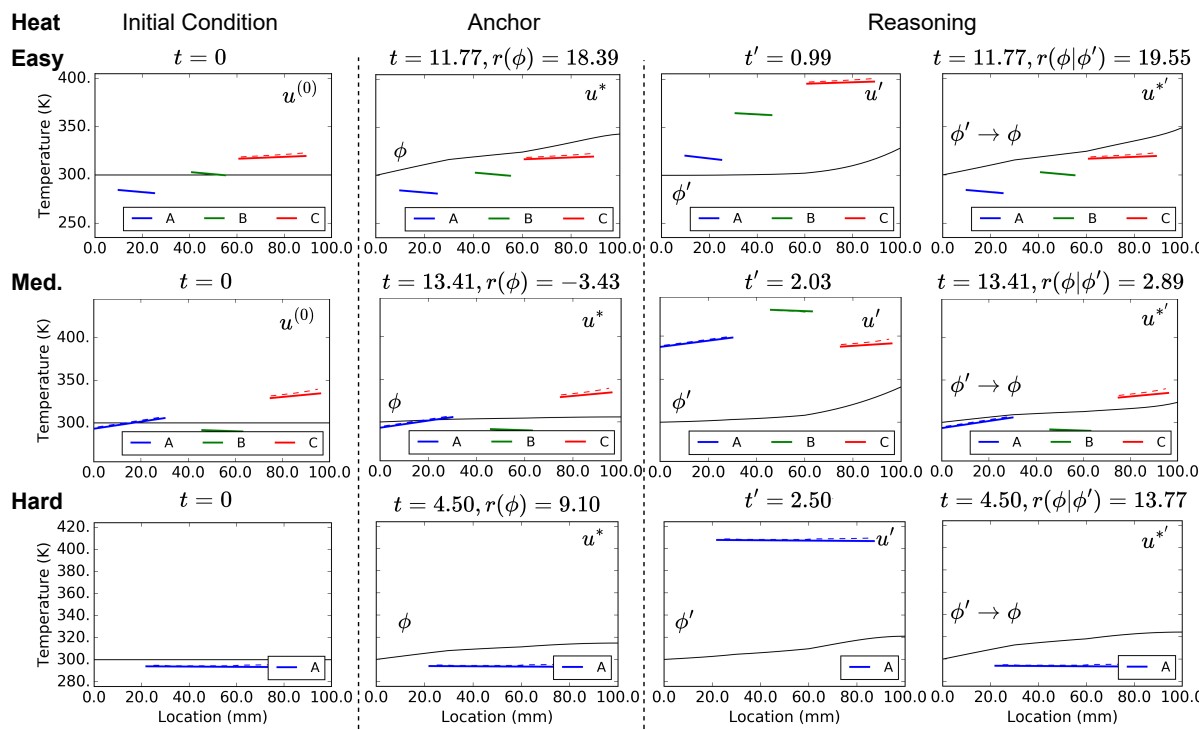

**Figure 7:** Case study of **heat** problems with different difficulty levels: easy (top), medium (middle), hard (bottom). Symbols are aligned with Fig. 5. From left to right: Directly solving $\phi$ from the initial condition $u^{(0)}$ (1st column) yields $r(\phi)$ (2nd column). Reasoning: solving $\phi'$ from $u^{(0)}$ to get $u'$ (3rd column) then solving $\phi$ from $u'$ to get $r(\phi|\phi')$ (4th column) Black curves indicate the system's states (temperature for heat, displacement for wave) at $t$ or $t'$. Colored segments are constraints, with dashes for inequalities ($\geq$ when dashes are above the solid, $\leq$ when dashes are below the solid). Note that although we always plot constraints (segments), *they actually constrain different temporal ranges* $[t_1, t_2]$ and it is possible that $t, t' \notin [t_1, t_2]$.

# B. Case Examples and Visualizations

To better illustrate our PDE reasoning, we show more cases with visualizations and their corresponding STLs (constraints). All time constraints are rounded to two decimal places, and parameters that describe the linear profiles are rounded to four decimal places.

### B.1. Heat

We show easy/medium/hard problems in Fig. 7, with their anchor STL ($\phi$) and subgoal STL ($\phi'$) listed below.

**1) Easy:**

Anchor Constraints STL ($\phi$):

$$F_{[1.17,3.48]}(\forall x \in [10, 25](u(x) - (-0.2169 \cdot x + 286.5171) > 0)) \wedge$$
$$(G_{[4.64,5.13]}(\forall x \in [41, 55](u(x) - (-0.2225 \cdot x + 311.8826) < 0)) \vee$$
$$F_{[6.04,11.77]}(\forall x \in [61, 89](u(x) - (0.0988 \cdot x + 310.7904) > 0)))$$

Subgoal STL Proposal ($\phi'$) by Controller:

$$F_{[0.42,0.99]}(\forall x \in [10, 25](u(x) - (-0.2907 \cdot x + 323.3970) > 0)) \wedge$$
$$(G_{[0.10,0.57]}(\forall x \in [31, 46](u(x) - (-0.1338 \cdot x + 368.9958) < 0)) \vee$$
$$F_{[0.30,0.96]}(\forall x \in [61, 89](u(x) - (0.0788 \cdot x + 390.7948) > 0)))$$

**2) Medium:**

Anchor Constraints STL ($\phi$):

$$G_{[2.18,2.70]}(\forall x \in [0, 30](u(x) - (0.4159 \cdot x + 293.2549) > 0)) \vee$$
$$(G_{[4.03,7.79]}(\forall x \in [46, 63](u(x) - (-0.0956 \cdot x + 296.0596) < 0)) \wedge$$
$$F_{[8.33,13.41]}(\forall x \in [75, 96](u(x) - (0.2602 \cdot x + 309.7111) > 0)))$$

Subgoal STL Proposal ($\phi'$) by Controller:

$$G_{[0.66,1.68]}(\forall x \in [0, 30](u(x) - (0.3616 \cdot x + 387.4454) > 0)) \vee$$
$$(G_{[0.52,2.03]}(\forall x \in [46, 63](u(x) - (-0.1061 \cdot x + 435.4267) < 0)) \wedge$$
$$F_{[0.22,1.17]}(\forall x \in [75, 96](u(x) - (0.1779 \cdot x + 374.4556) > 0)))$$

**3) Hard:**

Anchor Constraints STL ($\phi$):

$$G_{[2.62,4.50]}(\forall x \in [22, 87](u(x) - (-0.0122 \cdot x + 294.2976) > 0))$$

Subgoal STL Proposal ($\phi'$) by Controller:

$$G_{[1.29,2.50]}(\forall x \in [22, 87](u(x) - (-0.0157 \cdot x + 408.1535) > 0))$$

**B.2. Wave**

We show easy/medium/hard problems in Fig. 8, with their anchor STL ($\phi$) and subgoal STL ($\phi'$) listed below.

**1) Easy:**

Anchor Constraints STL ($\phi$):

$$(G_{[0.25,0.54]}(\forall x \in [7207, 23479](u(x) - (2.2684e - 05 \cdot x + 1.4129) < 0)) \wedge$$
$$F_{[0.76,0.84]}(\forall x \in [42469, 65095](u(x) - (1.8952e - 06 \cdot x - 1.7928) > 0))) \vee$$
$$F_{[1.12,1.33]}(\forall x \in [77653, 85444](u(x) - (-4.0675e - 05 \cdot x + 2.1560) > 0))$$

Subgoal STL Proposal ($\phi'$) by Controller:

$$(G_{[0.10,0.24]}(\forall x \in [7207, 23479](u(x) - (3.0242e - 05 \cdot x + 1.1961) < 0)) \wedge$$
$$F_{[0.05,0.09]}(\forall x \in [42469, 65095](u(x) - (2.3575e - 06 \cdot x - 1.5473) > 0))) \vee$$
$$F_{[0.10,0.18]}(\forall x \in [77653, 85444](u(x) - (-4.4208e - 05 \cdot x + 2.7018) > 0))$$

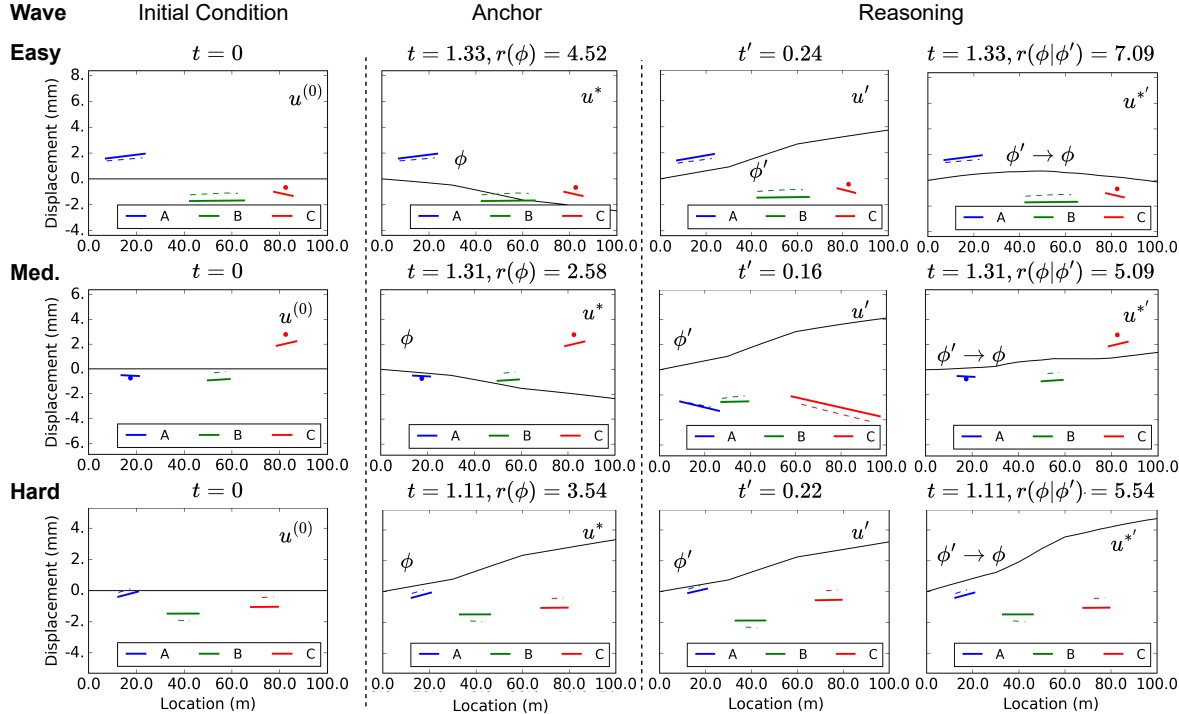

**Figure 8:** Case study of **wave** problems with different difficulty levels: easy (top), medium (middle), hard (bottom). Symbols are aligned with Fig. 5. From left to right: Directly solving $\phi$ from the initial condition $u^{(0)}$ (1st column) yields $r(\phi)$ (2nd column). Reasoning: solving $\phi'$ from $u^{(0)}$ to get $u'$ (3rd column) then solving $\phi$ from $u'$ to get $r(\phi|\phi')$ (4th column) Black curves indicate the system's states (temperature for heat, displacement for wave) at $t$ or $t'$. Colored segments are constraints, with dashes for inequalities ($\geq$ when dashes are above the solid, $\leq$ when dashes are below the solid). Note that although we always plot constraints (segments), *they actually constrain different temporal ranges* $[t_1, t_2]$ and it is possible that $t, t' \notin [t_1, t_2]$.

**2) Medium:**

Anchor Constraints STL ($\phi$):

$$G_{[0.10,0.24]}(\forall x \in [13787, 21080](u(x) - (-9.5400e - 06 \cdot x - 0.3744) < 0)) \vee$$
$$F_{[0.05,0.09]}(\forall x \in [49923, 59039](u(x) - (1.2003e - 05 \cdot x - 1.5231) > 0)) \vee$$
$$G_{[0.78,1.31]}(\forall x \in [78762, 86964](u(x) - (4.3983e - 05 \cdot x - 1.5994) > 0))$$

Subgoal STL Proposal ($\phi'$) by Controller:

$$G_{[0.06,0.16]}(\forall x \in [9084, 26246](u(x) - (-4.3491e - 05 \cdot x - 2.1348) > 0)) \wedge$$
$$G_{[0.04,0.04]}(\forall x \in [27204, 39168](u(x) - (4.2688e - 06 \cdot x - 2.6843) > 0)) \wedge$$
$$G_{[0.01,0.07]}(\forall x \in [58194, 97070](u(x) - (-4.0965e - 05 \cdot x + 0.2641) < 0))$$

**3) Hard:**

Anchor Constraints STL ($\phi$):

$$(G_{[0.23,0.30]}(\forall x \in [12400, 20684](u(x) - (4.0369e - 05 \cdot x - 0.9002) > 0)) \wedge$$
$$F_{[0.72,0.82]}(\forall x \in [33059, 46052](u(x) - (3.5491e - 07 \cdot x - 1.4933) < 0))) \vee$$
$$F_{[1.10,1.11]}(\forall x \in [67963, 79313](u(x) - (1.6090e - 06 \cdot x - 1.1675) > 0))$$

Subgoal STL Proposal ($\phi'$) by Controller:

$$(G_{[0.07,0.22]}(\forall x \in [12400, 20684](u(x) - (3.4256e - 05 \cdot x - 0.5172) > 0))\wedge$$
$$F_{[0.04,0.11]}(\forall x \in [33059, 46052](u(x) - (2.9472e - 07 \cdot x - 1.8896) < 0)))\vee$$
$$F_{[0.01,0.20]}(\forall x \in [67963, 79313](u(x) - (2.3060e - 06 \cdot x - 0.7148) > 0))$$

### B.3. STLs for Examples in Fig. 6

**Heat:**

Anchor Constraints STL ($\phi$) shown as A, B, C respectively in the *Initial Condition, Anchor* and right *Reasoning* columns of Fig. 6:

$$G_{[1.63,3.13]}(\forall x \in [2, 29](u(x) - (0.4565 \cdot x + 287.7909) > 0))\vee$$
$$(G_{[4.25,4.58]}(\forall x \in [38, 47](u(x) - (0.2137 \cdot x + 287.1038) < 0))\wedge$$
$$F_{[5.94,9.86]}(\forall x \in [54, 77](u(x) - (0.3008 \cdot x + 299.3877) > 0)))$$

Subgoal STL Proposal ($\phi'$) by Controller shown as A, B, C respectively in the left *Reasoning* column of Fig. 6:

$$G_{[0.69,1.49]}(\forall x \in [2, 29](u(x) - (0.4088 \cdot x + 294.5123) > 0))\vee$$
$$(G_{[0.26,0.33]}(\forall x \in [38, 47](u(x) - (0.2907 \cdot x + 404.7615) < 0))\wedge$$
$$F_{[0.06,0.10]}(\forall x \in [54, 77](u(x) - (0.3503 \cdot x + 316.1354) > 0)))$$

**Wave:**

Anchor Constraints STL ($\phi$) shown as A, B, C respectively in the *Initial Condition, Anchor* and right *Reasoning* columns of Fig. 6:

$$(G_{[0.16,0.20]}(\forall x \in [14057, 29980](u(x) - (2.8994e - 05 \cdot x - 2.5372) > 0))\wedge$$
$$G_{[0.28,0.37]}(\forall x \in [38096, 58208](u(x) - (-2.9597e - 05 \cdot x - 0.8070) > 0)))\vee$$
$$F_{[0.45,0.92]}(\forall x \in [71793, 88339](u(x) - (1.2523e - 05 \cdot x - 2.4337) < 0))$$

Subgoal STL Proposal ($\phi'$) by Controller shown as A, B, C respectively in the left *Reasoning* column of Fig. 6:

$$(G_{[0.00,0.01]}(\forall x \in [14057, 29980](u(x) - (3.0385e - 05 \cdot x - 1.3785) > 0))\wedge$$
$$G_{[0.03,0.07]}(\forall x \in [38096, 58208](u(x) - (-2.2655e - 05 \cdot x - 0.5356) > 0)))\vee$$
$$F_{[0.00,0.11]}(\forall x \in [71793, 88339](u(x) - (1.6430e - 05 \cdot x - 1.7368) < 0))$$

### B.4. Solutions to Examples in Fig. 6

Fig.9 presents the optimized control inputs for the **heat** and **wave** problems shown in Fig.6. The figure illustrates how the heat source ($q_t$) and the force ($F_t$) applied at the end of the rod change over time. We provide the solutions for 2 cases, the first column shows the solution obtained by directly solving the problem $\phi$, while the second column shows the solution to solve the problem $\phi$ after the subgoal $\phi'$.

## C. Dataset Details

### C.1. PDE Parameter Ranges

For generating our dataset, we sample hyperparameters (that define control problems) in the following range (Table 12 and 13). Note that our heat and wave problems are all 1D, making some dimension-related units different from those in 3D.

### C.2. Rules for Data Generation

**From STL to Natural Language.** Each natural language problem consists of two parts: one part defines the premises, such as the material density, initial temperature, and rod length; the other part describes the constraints, which can be expressed as an STL formula.

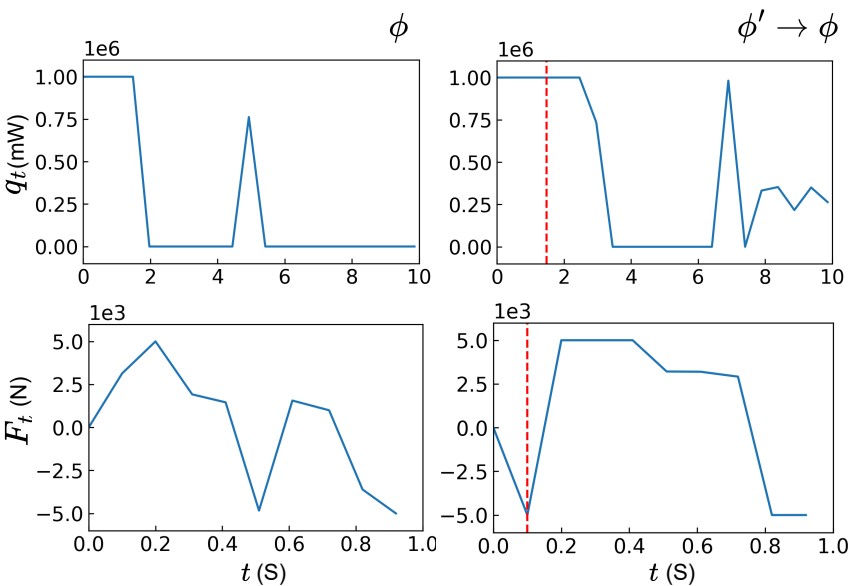

**Figure 9:** Synthesized control inputs for the **heat** (top) and the **wave** (bottom) problems in Fig.6. Left: Solution for directly solving $\phi$. Right: Solution for solving $\phi$ based on subgoal $\phi'$, where red vertical dashes indicate the control shift from $\phi'$ to $\phi$.

**Table 12:** Ranges for hyperparameters used in heat problems.

| Rod Length (mm) | Fixed Temp. (K) | Max Time (s) | Linear Profile Param. | | Thermal Conductivity ($mW \cdot mm/K$) | | Density ($kg/mm$) | | Specific Heat Capacity ($\mu J/kg/K$) | |
|---|---|---|---|---|---|---|---|---|---|---|
| $L$ | $temp$ | $t_{\max}$ | $a$ | $b$ | $\kappa_a$ ($\times 10^6$) | $\kappa_b$ ($\times 10^6$) | $\rho_a$ ($\times 10^{-6}$) | $\rho_b$ ($\times 10^{-6}$) | $c_a$ ($\times 10^8$) | $c_b$ ($\times 10^8$) |
| [50, 300] | [250, 350] | [5, 15] | [-0.5, 0.5] | $temp$ + [-20, 20] | [1.2, 1.8] | [0.4, 1.2] | [3, 6] | [3, 6] | [3, 4.5] | [4.5, 4.8] |

**Table 13:** Ranges for hyperparameters used in wave problems.

| Rod Length (mm) | Density ($kg/mm$) | | Young's Modulus ($N$) | | Max Time (s) | Linear Profile Param. | |
|---|---|---|---|---|---|---|---|
| $L$ | $\rho_{\text{steel}}$ ($\times 10^{-6}$) | $\rho_{\text{brass}}$ ($\times 10^{-6}$) | $E_{\text{steel}}$ ($\times 10^8$) | $E_{\text{brass}}$ ($\times 10^8$) | $t_{\max}$ | $a$ ($\times 10^{-5}$) | $b$ |
| [60000, 140000] | [7.6, 8] | [8.4, 8.8] | [2, 2.4] | [1, 1.8] | [0.5, 2] | [-5, 5] | [-3, 3] |

To convert the STL formula into informal language, each constraint and condition is mapped to corresponding phrases. For instance, the constraint conditions **F** and **G** are described as "for one point during" and "for all time between", respectively. The comparison conditions are mapped based on the problem type: for heat problems, $>$ indicates "the temperature distribution of the rod should be *greater than*"; for wave problems, $>$ indicates "the displacement of the rod should be *stretched over*".

We then consider the $\lor$ and $\land$ logical operators after converting each individual constraint. For problems with two constraints, we introduce transition words such as "moreover" for $\land$ and "either... or..." for $\lor$. For problems with three constraints, we design templates that account for the hierarchy of constraints based on the placement of parentheses in the STL formula that defines the logical order.

For example, given the STL syntax $(\mathbf{A} \land \mathbf{B}) \lor \mathbf{C}$, the template is: "Either satisfy the conditions that **A** and also **B**; or satisfy the condition that **C**." For $\mathbf{A} \land (\mathbf{B} \lor \mathbf{C})$, the template is: "Satisfy **A**. Afterwards, either consider **B** or **C**."

**From STL to Python.** To parse the predicted STL into Python in Table 9, we first extract logical connectives ($\lor, \land$), rod intersections, and constraint equations. Based on the number of constraints, the corresponding variables are inserted into a Python script template. The time intervals and constraint conditions ($\mathbf{G}, \mathbf{F}$) are passed directly into the script to preserve the original STL syntax. The generated outputs may fail to convert due to hallucinations or syntax errors.

For example, given the **STL Logic** below:

$$\mathbf{G}_{[0.049,0.053]}(\forall x \in [9829, 19907](u(x) - (1.882e - 05 \cdot x + 0.187) < 0)) \wedge$$
$$\mathbf{F}_{[0.051,0.149]}(\forall x \in [40199, 56082](u(x) - (3.356e - 06 \cdot x + -0.510) < 0))) \vee$$
$$\mathbf{F}_{[0.061,0.169]}(\forall x \in [75646, 98769](u(x) - (-1.390e - 05 \cdot x + 2.844) > 0)$$

We convert into its **Python snippet**:

```
apc0 = logic.APCont([9829, 19907], "<",
        lambda x: 1.882e-05 * x + 0.187, lambda x: 1.882e-05)
apc1 = logic.APCont([40199, 56082], "<",
        lambda x: 3.356e-06 * x + -0.510, lambda x: 3.356e-06)
apc2 = logic.APCont([75646, 98769], ">",
        lambda x: -1.390e-05 * x + 2.844, lambda x: -1.390e-05)
cregions = {"A": apc0,"B": apc1,"C": apc2}
cspec = "(((G_[0.049, 0.053] (A)) &
        (F_[0.051, 0.149] (B))) |
        (F_[0.061, 0.169] (C)))"
```

### C.3. ChatGPT Augmentation

We adopt the data augmentation pipeline proposed by (Pan et al., 2023), where ChatGPT 4o-mini is prompted to rephrase each natural language problem five times. The problems are formulated based on STL logic with PDE parameter variables enclosed in squared brackets, which are further replaced with randomly values sampled from ranges in Appendix C.1. For example:

**Prompt**: Rephrase the source sentence in 5 different ways. Make the outputs as diverse as possible. Make sure not to change the equations starting with mu and parameters in the squared brackets, and maintain all the parameters while rephrasing.

**Source (original problem)**: For one point during the time interval [t0] and [t1], the temperature distribution of the rod should be larger than the linear profile mu0(x) = [a0] * x + [b0] between section [r0] and [r1].

**GPT Outputs**:

1. At a specific moment within the time period [t0] to [t1], the temperature distribution along the rod must exceed the linear profile mu0(x) = [a0] * x + [b0] in the segment defined by [r0] and [r1].

2. Within the time frame of [t0] to [t1], there should be at least one instance where the rod's temperature distribution surpasses the linear profile mu0(x) = [a0] * x + [b0] from [r0] to [r1].

3. During the interval between [t0] and [t1], there is a moment at which the temperature across the rod must be greater than the linear profile mu0(x) = [a0] * x + [b0] within the range of [r0] and [r1].

4. Throughout the duration from [t0] to [t1], there exists a point in time where the temperature distribution of the rod should be greater than the linear profile mu0(x) = [a0] * x + [b0] applicable between the sections [r0] and [r1].

5. At some time during the interval between [t0] and [t1], it is required that the temperature distribution of the rod exceeds the linear profile mu0(x) = [a0] * x + [b0] in the region from [r0] to [r1].

Table 14 shows the character-level relative edit distance between each natural language problem and its 5 augmentations based on the problem type and the number of constraints.

## D. Collection of Manually Written PDE Control Problems

We overview our questionnaire in Fig. 10. For the questionnaire collection, each Zoom session takes one hour, including background instruction and manual design (with interactive guidance). Although it is challenging to make a fair statistical comparison with human experts in terms of performance (both time cost and accuracy), our LLMs (autoformalization → Controller → code generation) complete the task in under 120 seconds on a single 6000 Ada GPU. This is significantly faster than the human volunteers we recruited (for collecting our real-world samples), who required several minutes just to read the original informal problem. We further show the statistics of the background of participants in Fig. 11.

**Table 14:** Averaged relative edit distance between the problem's natural language and 5 GPT augmentations. Deviations are in parentheses.

| | Relative Edit Distance | | |
|---|---|---|---|
| PDE | Number of Constraints | | |
| | 1 | 2 | 3 |
| Heat | 0.431 (0.0545) | 0.456 (0.0505) | 0.490 (0.0398) |
| Wave | 0.455 (0.0524) | 0.472 (0.0433) | 0.484 (0.0352) |

**Difference between Synthetic and Manual Data**  We observe some differences between synthetic and manually generated data that may lead to the model's failure to produce valid STL logic:

- Ambiguous symbol usage, such as using "ho" instead of "rho" to denote material density.

- Inconsistent units within a single sentence. For example, "Assume that the discretized time interval is 0.05s and the maximum time is 7400 milliseconds."

- Insufficient information, where four samples fail to fully describe the material properties. For instance, the manual data only provides the density of one material after the statement "the rod is composed of two materials".

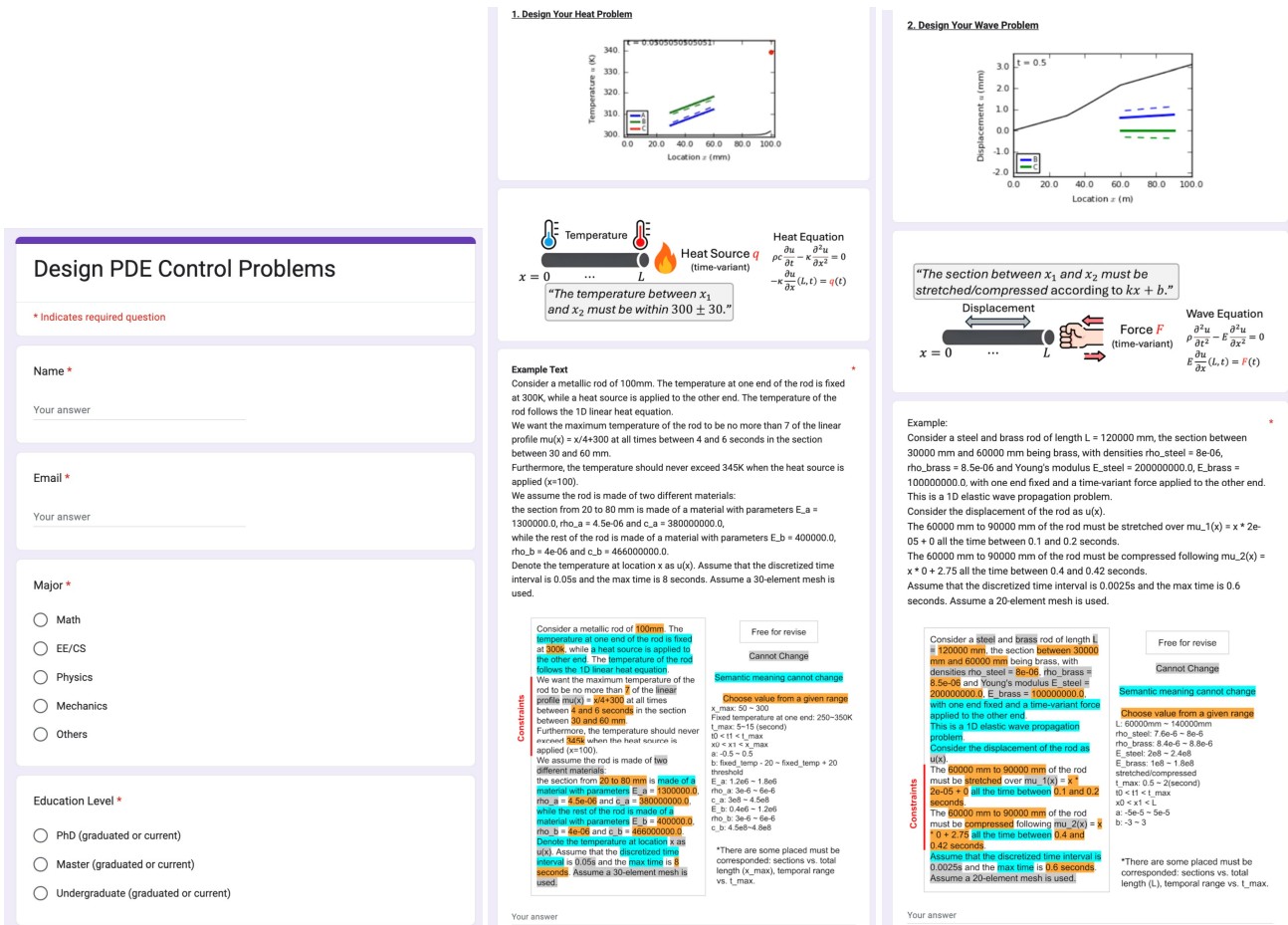

**Figure 10:** Google Form for collecting manually written PDE control problems.

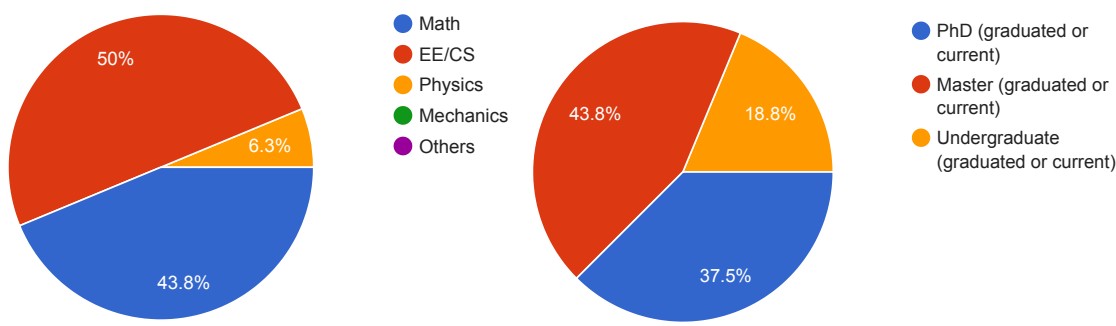

**Figure 11:** Background of our questionnaire participants.

## E. Training Details

We leverage the pretrained MathCoder2-DeepSeekMath-7B (Lu et al., 2024) checkpoint (MathCoder2) which has a 4096-token context length. All our *trained* models are evaluated zero-shot. For fair comparison of our models with MathCoder2, we provide two few-shot examples for the latter to leave sufficient tokens to generate a valid output (whether STL or Python). We similarly provide two few-shot examples for GPT-4o and o1-mini.

Our instructions are structured in the Alpaca format (Taori et al., 2023).

### E.1. Autoformalization: SFT of Translator

The Translator was trained with two 6000Ada GPUs using a per-GPU batch size of 16 and 4 gradient accumulation steps for a total of 3000 steps. We fine-tuned the MathCoder2-DeepSeekMath-7B parameter model from (Lu et al., 2024) with LoRA rank $r = 64$ and $\alpha = 256$.

**Prompt:** Below is a natural language description of partial differential equation optimization problem. Translate the problem into Latex code following spatial-signal temporal logic.

### E.2. Program Synthesis: SFT of Coder

The Coder further fine-tuned the Translator with supervised fine-tuning and LoRA, rank $r = 64$ and $\alpha = 256$, to produce Python code from natural language and STL pairs. This was trained with two 6000Ada GPUs using a per-GPU batch size of 8 and 8 gradient accumulation steps for a total of 3000 steps.

We design two prompts, firstly for generating python code aligned with the anchor STL, and secondly for generating python code aligned with the proposed subgoal STL. The natural language problem is provided in both cases to extract system settings.

**Prompt:** Below is a natural language description of partial differential equation optimization problem, paired with your spatial-signal temporal logic description of the same problem provided earlier. Note that there may be mistakes in the spatial-signal temporal logic statement but the natural language description is accurate. Translate the problem into Python code following spatial-signal temporal logic.

**Prompt:** Below is a natural language description of partial differential equation optimization problem, paired with your spatial-signal temporal logic description of an intermediate problem provided earlier. Instead of optimizing the natural language problem directly, we want to optimize the intermediate problem to produce a state that will better serve to achieve the final conditions outlined in the natural language problem. Your spatial-signal temporal logic description in latex paired to the original problem describes this intermediate problem. Translate the intermediate problem into Python code following spatial-signal temporal logic.

In practice, we find it to be helpful to supervise fine-tune the Coder with misaligned (NL, STL) pairs to promote the subgoal STL constraints when synthesizing the corresponding Python program for the subgoal STL.

### E.3. Reasoning: RLHF of Controller

The Controller is trained with DPO (Rafailov et al., 2024) from the Translator checkpoint with LoRA rank $r = 64$ and $\alpha = 256$. We train with two 6000Ada GPUs using a per-GPU batch size of 2 and 4 gradient accumulation steps for a total of 16,800 steps. For DPO, we set $\beta = 0.1$, and $\lambda = 1$ in Eq. 3.

**Prompt:** Below is a natural language description of partial differential equation optimization problem. Instead of optimizing the provided problem directly, we want to optimize an intermediate problem to produce a state that will better serve to achieve the final conditions outlined in the natural language problem. Generate a spatial-signal temporal logic description in Latex code for such an intermediate problem.

## F. More Experiment Results

### F.1. End-to-End Evaluation of Program Synthesis

In Table 15 and Table 16, we provide end-to-end results of program synthesis on synthetic and manual data, respectively. In these results, coders take LLM-generated (noisy) STLs in their prompt. Overall, our Coder LLM still achieves strong results. Moreover, we also show another baseline "Coder-only", where the Coder LLM takes only natural language as the input without explicitly formalized STLs. In Table 16, we can see that our method (autoformalization + program synthesis) outperforms "Coder-only" (direct program synthesis without autoformalization), emphasizing the importance of leveraging formal language (STL).

**Table 15:** End-to-end autoformalization and program synthesis. The Coder produces Python using the Translator's STL output. MathCoder2 produces Python using its own STL output. Coder-only is a MathCoder2 model fine-tuned to produce python directly from natural language and seeing no STL. To be comparable with the Translator + Coder autoformalization and program synthesis, Coder-only is trained for 6000 steps with the same settings as the Translator and Coder (Appendix E). The evaluation for Coders is zero-shot. MathCoder2 is evaluated with two few-shot examples. Deviations over 3 seeds are in parentheses. **Bold** indicates the best, underline indicates the runner up.

| PDE | Model | Executability (↑) (Coder) | **Utility** **RMSE** (↓) |
|---|---|---|---|
| | Ours | **0.9978** (0.0015) | **0.0174** (0.0065) |
| Heat | MathCoder2 | 0.9197 (0.02731) | 1.3841 (0.1196) |
| | Ours (Coder-only) | **0.9978** (0.0009) | 0.0235 (0.00163) |
| | Ours | 0.9620 (0.0098) | **0.0076** (0.0011) |
| Wave | MathCoder2 | 0.8305 (0.0475) | 0.8332 (0.0965) |
| | Ours (Coder-only) | **0.9779** (0.002148) | 0.02522 (0.000583) |

**Table 16:** End-to-end autoformalization and program synthesis on manually written data (Sec. 3.2). The Coder produces Python using the Translator's STL output. MathCoder2 produces Python using its own STL output. Coder-only is a MathCoder2 model fine-tuned to produce python directly from natural language and seeing no STL. To be comparable with the Translator + Coder autoformalization and program synthesis, Coder-only is trained for 6000 steps with the same settings as the Translator and Coder (Appendix E). The evaluation for Coders is zero-shot. MathCoder2 is evaluated with two few-shot examples. Deviations over 3 seeds are in parentheses. **Bold** indicates the best, underline indicates the runner up.

| PDE | Model | Executability (↑) (Coder) | **Utility** **RMSE** (↓) |
|---|---|---|---|
| | Ours | 0.4510 (0.0832) | **0.1837** (0.0095) |
| Heat | MathCoder2 | 0.4902 (0.1386) | 0.2928 (0.0780) |
| | Ours (Coder-only) | **0.6078** (0.0555) | 2.442 (0.0) |
| | Ours | **1.0** (0.0) | **0.0119** (0.0) |
| Wave | MathCoder2 | 0.9020 (0.0277) | 1.767 (0.8109) |
| | Ours (Coder-only) | 0.4706 (0.0) | 0.0890 (0.0) |

### F.2. PDE Reasoning

#### F.2.1. End-to-End Evaluation of PDE Reasoning

We include end-to-end evaluation results, where we use the Coder LLM to generate Python code after the Controller LLM propose subgoal STLs in Table 17. In general, our Controller still shows strong reasoning capability (both the success rate

**Table 17:** End-to-end scientific reasoning over PDE control problems via our Controller LLM. Deviations over 5 seeds are in parentheses. "Valid STL $\phi'$ (%)" is the ratio of proposed subgoal STL $\phi'$ without any syntax errors. **Bold** indicates the best, underline denotes the runner-up. "x" indicates no valid STLs were generated for evaluation. "-" indicates not applicable.

| PDE | Difficulty Level | Success Rate $\overline{P}\uparrow$ | | | | Utility Gain $\overline{\Delta r}\uparrow$ | | | |
|---|---|---|---|---|---|---|---|---|---|
| | | Ours | Math-Coderv2 | GPT (o1-mini) | GPT (4o) | Ours | Math-Coderv2 | GPT (o1-mini) | GPT (4o) |
| Heat | Easy | **0.490** (0.1666) | 0.399 (0.2605) | 0.154 (0) | 0.160 (0.0686) | **0.651** (1.1150) | 0.352 (1.3309) | -1.262 (0) | -0.865 (2.0310) |
| | Medium | **0.345** (0.1640) | 0.339 (0.2345) | 0 (0) | 0.121 (0.0497) | -1.055 (0.9657) | **-0.061** (0.4098) | -2.455 (0) | -2.294 (1.7773) |
| | Hard | **0.307** (0.1363) | 0.239 (0.1895) | 0.074 (0.0262) | 0.150 (0.0797) | -1.788 (2.1369) | **-1.254** (0.8047) | -3.992 (0.5175) | -2.603 (2.4486) |
| | All | **0.381** (0.1556) | 0.326 (0.2282) | x | 0.144 (0.0660) | -0.731 (1.4059) | **-0.321** (0.8485) | x | -1.921 (2.0856) |
| Wave | Easy | 0.897 (0.0502) | 0.854 (0.0894) | x | **1** (0) | 1.423 (0.4135) | 1.173 (0.6424) | x | **1.907** (0.0333) |
| | Medium | 0.836 (0.0936) | 0.743 (0.0410) | x | **0.958** (0.0105) | **0.865** (0.4508) | 0.758 (0.4265) | x | 0.803 (0.2279) |
| | Hard | 0.331 (0.1214) | 0.336 (0.1041) | x | **0.416** (0.1113) | **-0.289** (0.5723) | -0.649 (0.3948) | x | -0.376 (0.2142) |
| | All | 0.688 (0.0884) | 0.644 (0.0781) | x | **0.791** (0.0406) | 0.698 (0.5773) | 0.427 (0.4879) | x | **0.778** (0.1585) |
| Valid STL $\phi'$ (%) (↑) | | **82.70** (1.97) | 42.45 (10.54) | 0.04 (0.10) | 2.55 (0.65) | **82.70** (1.97) | 42.45 (10.54) | 0.04 (0.10) | 2.55 (0.65) |
| Valid Python Code (%) (↑) | | **75.65** (1.50) | 27.95 (5.72) | 0.09 (0.78) | 3.25 (0.63) | **75.65** (1.50) | 27.95 (5.72) | 0.09 (0.78) | 3.25 (0.63) |

**Table 18:** Proportion of valid subgoal STL generations by different controller models in Table 9 and 17 (in decimal form). Deviations over 5 seeds are in parentheses.

| PDE | Difficulty Level | Proportion ($\in [0,1]$) of valid STL $\phi'$ (↑) | | | |
|---|---|---|---|---|---|
| | | Ours | Math-Coderv2 | GPT (o1-mini) | GPT (4o) |
| Heat | Easy | 0.8851 (0.0110) | 0.507 (0.1354) | 0 | 0.028 (0.0078) |
| | Medium | 0.891 (0.0243) | 0.491 (0.0962) | 0.0004 (0.0015) | 0.031 (0.0046) |
| | Hard | 0.881 (0.0148) | 0.493 (0.0956) | 0.002 (0.0045) | 0.052 (0.0113) |
| | All | 0.886 (0.0167) | 0.497 (0.1091) | 0.0008 (0.002) | 0.037 (0.0079) |
| Wave | Easy | 0.756 (0.0228) | 0.309 (0.1113) | 0 | 0.008 (0.0066) |
| | Medium | 0.748 (0.0257) | 0.397 (0.0947) | 0 | 0.015 (0.0052) |
| | Hard | 0.799 (0.0197) | 0.349 (0.0990) | 0 | 0.018 (0.0036) |
| | All | 0.768 (0.0227) | 0.352 (0.1016) | 0 | 0.014 (0.0051) |

**Table 19:** Proportion of valid subgoal Python program generations by different coder models in Table 17 (in decimal form). Deviations over 5 seeds are in parentheses.

| PDE | Difficulty Level | Proportion ($\in [0,1]$) of valid Python program (↑) | | | |
|---|---|---|---|---|---|
| | | Ours | Math-Coderv2 | GPT (o1-mini) | GPT (4o) |
| Heat | Easy | 0.754 (0.0244) | 0.301 (0.0624) | 0.014 (0.0193) | 0.047 (0.0113) |
| | Medium | 0.767 (0.0261) | 0.295 (0.0503) | 0.012 (0.0124) | 0.039 (0.0048) |
| | Hard | 0.828 (0.0113) | 0.329 (0.0532) | 0.027 (0.0147) | 0.058 (0.0085) |
| | All | 0.783 (0.0206) | 0.308 (0.0553) | 0.018 (0.0155) | 0.048 (0.0082) |
| Wave | Easy | 0.702 (0.0080) | 0.211 (0.0675) | 0 | 0.012 (0.0048) |
| | Medium | 0.713 (0.0113) | 0.278 (0.0538) | 0 | 0.018 (0.0041) |
| | Hard | 0.776 (0.0086) | 0.265 (0.0561) | 0 | 0.021 (0.0044) |
| | All | 0.730 (0.0093) | 0.251 (0.0591) | 0 | 0.017 (0.0044) |

and utility gain), and also high rate of proposing valid subgoal STL ($\phi'$).

### F.2.2. PROPORTION OF VALID PREDICTIONS

We notice that the number of valid predictions varied significantly depending on the model. Therefore, we include Table 18 for STL and Table 19 for Python program to comprehensively show the number of valid predictions that each model makes under each type of problem and difficulty level. Please note that the values in Table 18 and Table 19 are not expressed as percentages.

# G. More Related Works

## G.1. Autoformalization in AI-for-math

Autoformalization, the process of converting informal mathematical statements or instructions into formal representations, is explored through a variety of techniques. A significant subset of works employed LLMs to translate informal descriptions into formal representations. VernaCopter (van de Laar et al., 2024) leveraged LLMs to convert natural language commands into Signal Temporal Logic (STL) specifications, integrating syntax and semantic checkers for correctness. Pretrained LLMs like GPT-3.5 and GPT-4o were leveraged to translate informal problems into Isabelle proof sketches, refining outputs through iterative prompting and heuristic-based validation (Zhou et al., 2024). Back-translation (Jiang et al., 2023) trained LLMs to map between informal and formal theorem statements in Lean4 and Isabelle. These approaches focused on leveraging LLMs for direct autoformalization while incorporating filtering mechanisms to improve reliability. In contrast, hybrid approaches interact between manual and autoformalization. Several studies combined expert-curated manual formalization with automated techniques to improve accuracy. DeepSeek-Prover (Xin et al., 2024), Trigo (Xiong et al., 2023), and (Murphy et al., 2024) used an iterative pipeline where initial formalization was manually crafted, followed by automated expansion and refinement. LILA (Mishra et al., 2022) similarly applied domain-specific rules and Python-based DSL annotations for automatic formalization while relying on human annotators for cases where automation failed. These hybrid approaches aimed to balance the scalability of automation with the precision of manual verification. In our work, we for the first time train LLMs to autoformalize informal PDE control problems into formalized STL logic.

## G.2. PDE Controls

PDEs are essential for modeling physical phenomena, helping researchers predict behaviors, optimize processes, and drive innovation across fields like climate modeling and material design. PDE control focuses on manipulating system behaviors, ensuring stability in applications like robotics and reactors, while enabling systems to adapt to changing conditions for more sustainable solutions (Alvarez, 2020; Holl et al., 2020; Ramos et al., 2022; Mukherjee & Liu, 2023; Wei et al.; 2024b). Adjoint methods have been widely used for controlling physical systems governed by PDEs due to their accuracy, despite being computationally expensive (Lions, 1971; McNamara et al., 2004; Protas, 2008), while deep learning-based approaches, such as supervised learning with differentiable physics losses (Holl et al., 2020; Hwang et al., 2022), optimize control directly via backpropagation through time. (Mowlavi & Nabi, 2022; Barry-Straume et al., 2022) optimizes control problems and the PDE system state with physics-informed neural networks. Reinforcement learning (RL) (Farahmand et al., 2017; Pan et al., 2018; Rabault et al., 2019) optimizes control by treating signals as actions for sequential decision-making in fluid dynamics applications like drag reduction, heat transfer, and swimming (Garnier et al., 2021). Pretrained LLMs can enhance PDE surrogate modeling by integrating textual descriptions of system information–such as boundary conditions and the governing PDE–into a multimodal learning framework (Lorsung & Farimani, 2025). PDE control can also be discretized and formulated into a mixed-integer linear programming (MILP) problem via finite element method (FEM) (Sadraddini & Belta, 2015; Alvarez, 2020). More recently, diffusion-based generation has been leveraged to jointly optimize the PDE simulations and control signals (Wei et al., 2024a). Different from traditional PDE control objectives such as setpoint/trajectory tracking and disturbance rejection (Paunonen & Phan, 2019; Paunonen & Humaloja, 2022), our utility score objective (Appendix A.2), which mainly aims to reduce distance to the target, can better handle inequality constraints compared to tracking errors. From an optimization perspective, our work leverages LLMs to propose better initializations (initial conditions) to solve the open-loop PDE control problem. Extensions to closed-loop control are possible by appending the utility from the LLM-proposed optimization into future optimization rounds. This increases the complexity of LLM fine-tuning; thus, as the first step in this direction we focus on open-loop control.

## G.3. LLM-based Task Planning

Natural language (NL)-based task planning in robotics has gained increasing attention. Approaches such as (Pan et al., 2023) enable task-specific translations from informal language to Linear Temporal Logic (LTL), allowing robots to follow structured plans even in low-resource scenarios. Building on this foundation, recent research has explored the use of LLMs for task planning, demonstrating models' potential in decision-making and executing complex plans (Singh et al., 2023; Shah et al., 2023; Li et al., 2022). For instance, (Wang et al.) leveraged LLMs to arrange and predict execution sequences for robots, achieving a comparable success rate to human users. Additionally, (Ren et al., 2023) addressed the hallucination issue from LLM-based planners by incorporating uncertainty alignment, improving the reliability of generated plans. More recently, CLMASP (Lin et al., 2024) refined LLM-generated skeleton plans using Answer Set Programming (ASP) for robotic task execution. Inspired by these subgoal approaches, we for the first time train LLMs to perform reasoning and planning on PDE control problems.

# H. Datasheets for Datasets

This document is based on *Datasheets for Datasets* by (Gebru et al., 2021).

## H.1. Motivation

**For what purpose was the dataset created?** Was there a specific task in mind? Was there a specific gap that needed to be filled? Please provide a description.

The dataset was created to enable large language models (LLMs) to tackle complex Partial Differential Equation (PDE) control problems. The specific purpose is to advance automated formalization and reasoning in applied mathematics, addressing the lack of datasets tailored to PDE-related tasks. The dataset bridges informal natural language problems and formal specifications/code for PDE systems, fostering research in scientific computing and engineering.

**Who created this dataset (e.g., which team, research group) and on behalf of which entity (e.g., company, institution, organization)?**

The dataset was created by the anonymous authors of this PDE-Controller paper, affiliated with a research group focused on AI-for-math applications.

**What support was needed to make this dataset?** (e.g.who funded the creation of the dataset? If there is an associated grant, provide the name of the grantor and the grant name and number, or if it was supported by a company or government agency, give those details.)

The creation of the dataset was supported by research funding for developing novel applications of LLMs in applied mathematics. Further support included computational resources for fine-tuning LLMs and manual curation by domain experts.

**Any other comments?**

The dataset represents a pioneering effort to merge AI capabilities with PDE-based scientific reasoning, significantly expanding the potential applications of LLMs.

## H.2. Composition

**What do the instances that comprise the dataset represent (e.g., documents, photos, people, countries)?** Are there multiple types of instances (e.g., movies, users, and ratings; people and interactions between them; nodes and edges)? Please provide a description.

Each instance represents a PDE control problem, including: 1) Informal problem descriptions in natural language; 2) Formal specifications in Signal Temporal Logic (STL); 3) Python code that integrates PDE simulation and optimization tools.

**How many instances are there in total (of each type, if appropriate)?**

The dataset comprises over 2.13 million synthetic (natural language, STL, Python code) triplets, with additional real-world examples including 17 manually written heat problems and 17 wave problems.

**Does the dataset contain all possible instances or is it a sample (not necessarily random) of instances from a larger set?** If the dataset is a sample, then what is the larger set? Is the sample representative of the larger set (e.g., geographic coverage)? If so, please describe how this representativeness was validated/verified. If it is not representative of the larger set, please describe why not (e.g., to cover a more diverse range of instances, because instances were withheld or unavailable).

It is a synthesized dataset designed to cover a diverse range of PDE control problems, sampled and augmented from representative templates to ensure coverage of key scenarios and complexities.

**What data does each instance consist of?** "Raw" data (e.g., unprocessed text or images) or features? In either case, please provide a description.

Each instance includes: 1) Informal natural language descriptions of PDE problems; 2) Formal representations in STL syntax; 3) Python code for solving the PDE problem using optimizers such as Gurobi.

**Is there a label or target associated with each instance?** If so, please provide a description.
Yes, each instance includes ground-truth STL and Python code, verified for alignment and executability.

**Is any information missing from individual instances?** If so, please provide a description, explaining why this information is missing (e.g., because it was unavailable). This does not include intentionally removed information, but might include, e.g., redacted text.
Not Applicable.

**Are relationships between individual instances made explicit (e.g., users' movie ratings, social network links)?** If so, please describe how these relationships are made explicit.
Yes, relationships between natural language, STL specifications, and Python code are explicitly maintained for traceability.

**Are there recommended data splits (e.g., training, development/validation, testing)?** If so, please provide a description of these splits, explaining the rationale behind them.
Yes, the dataset is split into training and testing sets, with specific splits for heat and wave problems.

**Are there any errors, sources of noise, or redundancies in the dataset?** If so, please provide a description.
Synthetic data is validated through automated checks and human verification. Errors may arise from annotation inconsistencies, especially in manually curated problems.

**Is the dataset self-contained, or does it link to or otherwise rely on external resources (e.g., websites, tweets, other datasets)?** If it links to or relies on external resources, a) are there guarantees that they will exist, and remain constant, over time; b) are there official archival versions of the complete dataset (i.e., including the external resources as they existed at the time the dataset was created); c) are there any restrictions (e.g., licenses, fees) associated with any of the external resources that might apply to a future user? Please provide descriptions of all external resources and any restrictions associated with them, as well as links or other access points, as appropriate.
The dataset is self-contained, with no reliance on external or dynamic resources.

**Does the dataset contain data that might be considered confidential (e.g., data that is protected by legal privilege or by doctor-patient confidentiality, data that includes the content of individuals' non-public communications)?** If so, please provide a description.
No.

**Does the dataset contain data that, if viewed directly, might be offensive, insulting, threatening, or might otherwise cause anxiety?** If so, please describe why.
No.

**Does the dataset relate to people?** If not, you may skip the remaining questions in this section.
No.

**Does the dataset identify any subpopulations (e.g., by age, gender)?** If so, please describe how these subpopulations are identified and provide a description of their respective distributions within the dataset.
No.

**Is it possible to identify individuals (i.e., one or more natural persons), either directly or indirectly (i.e., in combination with other data) from the dataset?** If so, please describe how.

No.

**Does the dataset contain data that might be considered sensitive in any way (e.g., data that reveals racial or ethnic origins, sexual orientations, religious beliefs, political opinions or union memberships, or locations; financial or health data; biometric or genetic data; forms of government identification, such as social security numbers; criminal history)?** If so, please provide a description.
No.

**Any other comments?**
The dataset's richness in complexity and diversity makes it a significant resource for advancing applied mathematics via AI.

### H.3. Collection

**How was the data associated with each instance acquired?** Was the data directly observable (e.g., raw text, movie ratings), reported by subjects (e.g., survey responses), or indirectly inferred/derived from other data (e.g., part-of-speech tags, model-based guesses for age or language)? If data was reported by subjects or indirectly inferred/derived from other data, was the data validated/verified? If so, please describe how.
The data was synthesized from key PDE control templates, augmented through principled methods, and verified by experts. Real-world problems were collected via a questionnaire-based manual curation process involving students and researchers.

**Over what timeframe was the data collected?** Does this timeframe match the creation timeframe of the data associated with the instances (e.g., recent crawl of old news articles)? If not, please describe the timeframe in which the data associated with the instances was created. Finally, list when the dataset was first published.
The synthetic dataset was generated in late 2024, with real-world problems curated in 2025.

**What mechanisms or procedures were used to collect the data (e.g., hardware apparatus or sensor, manual human curation, software program, software API)?** How were these mechanisms or procedures validated?
Procedures included automated STL generation, natural language augmentation using GPT-4, and manual problem formulation.

**What was the resource cost of collecting the data?** (e.g. what were the required computational resources, and the associated financial costs, and energy consumption - estimate the carbon footprint.)
Resource costs included computational expenses for data synthesis and human time for manual curation and validation.

**If the dataset is a sample from a larger set, what was the sampling strategy (e.g., deterministic, probabilistic with specific sampling probabilities)?**
Not applicable.

**Who was involved in the data collection process (e.g., students, crowdworkers, contractors) and how were they compensated (e.g., how much were crowdworkers paid)?**
Graduate students and researchers with expertise in applied mathematics and AI.

**Were any ethical review processes conducted (e.g., by an institutional review board)?** If so, please provide a description of these review processes, including the outcomes, as well as a link or other access point to any supporting documentation.
No.

**Does the dataset relate to people?** If not, you may skip the remainder of the questions in this section.
No.

**Did you collect the data from the individuals in question directly, or obtain it via third parties or other sources (e.g., websites)?**
Our manually written data is collected from each individual with questions.

**Were the individuals in question notified about the data collection?** If so, please describe (or show with screenshots or other information) how notice was provided, and provide a link or other access point to, or otherwise reproduce, the exact language of the notification itself.
N/A

**Did the individuals in question consent to the collection and use of their data?** If so, please describe (or show with screenshots or other information) how consent was requested and provided, and provide a link or other access point to, or otherwise reproduce, the exact language to which the individuals consented.
N/A

**If consent was obtained, were the consenting individuals provided with a mechanism to revoke their consent in the future or for certain uses?** If so, please provide a description, as well as a link or other access point to the mechanism (if appropriate)
N/A

**Has an analysis of the potential impact of the dataset and its use on data subjects (e.g., a data protection impact analysis)been conducted?** If so, please provide a description of this analysis, including the outcomes, as well as a link or other access point to any supporting documentation.
No. Our data are intended to be used in evaluation only and all charts are publicly avi[able.

**Any other comments?** N/A

### H.4. Preprocessing / Cleaning / Labeling

**Was any preprocessing/cleaning/labeling of the data done(e.g.,discretization or bucketing, tokenization, part-of-speech tagging, SIFT feature extraction, removal of instances, processing of missing values)?** If so, please provide a description. If not, you may skip the remainder of the questions in this section.
Yes, preprocessing included: 1) Reformatting natural language prompts; 2) Validating STL and Python code for correctness and executability; 3) Augmenting natural language data using rephrasing techniques.

**Was the "raw" data saved in addition to the preprocessed/cleaned/labeled data (e.g., to support unanticipated future uses)?** If so, please provide a link or other access point to the "raw" data.
Yes, raw data and intermediate representations are retained for reproducibility and future use.

**Is the software used to preprocess/clean/label the instances available?** If so, please provide a link or other access point.
The tools and scripts for preprocessing are included in the supplementary materials of the PDE-Controller framework.

**Any other comments?**
Preprocessing ensures high-quality alignment between natural language, formal logic, and executable code.

### H.5. Uses

**Has the dataset been used for any tasks already?** If so, please provide a description.
Yes, it was used to train and evaluate the PDE-Controller framework and benchmark its performance against state-of-the-art LLMs.

**Is there a repository that links to any or all papers or systems that use the dataset?**  If so, please provide a link or other access point.
N/A

**What (other) tasks could the dataset be used for?**
The dataset could be used for: 1) Training models for scientific reasoning and formalization; 2) Developing tools for automated program synthesis; 3) Advancing research in AI-driven engineering and physics.

**Is there anything about the composition of the dataset or the way it was collected and preprocessed/cleaned/labeled that might impact future uses?**  For example, is there anything that a future user might need to know to avoid uses that could result in unfair treatment of individuals or groups (e.g., stereotyping, quality of service issues) or other undesirable harms (e.g., financial harms, legal risks) If so, please provide a description. Is there anything a future user could do to mitigate these undesirable harms?
N/A

**Are there tasks for which the dataset should not be used?**  If so, please provide a description.
The dataset is not suitable for tasks unrelated to PDE control or tasks requiring real-world human data.

**Any other comments?**
The dataset's structured format supports reproducible and extensible research in applied mathematics.

### H.6. Distribution

**Will the dataset be distributed to third parties outside of the entity (e.g., company, institution, organization) on behalf of which the dataset was created?**  If so, please provide a description.
Yes, the dataset will be made publicly available for research purposes.

**How will the dataset will be distributed (e.g., tarball on website, API, GitHub)?**  Does the dataset have a digital object identifier (DOI)?
The dataset will be distributed via GitHub and academic repositories, with accompanying documentation.

**When will the dataset be distributed?**
The dataset is expected to be released following the ICML 2025 conference.

**Will the dataset be distributed under a copyright or other intellectual property (IP) license, and/or under applicable terms of use (ToU)?**  If so, please describe this license and/or ToU, and provide a link or other access point to, or otherwise reproduce, any relevant licensing terms or ToU, as well as any fees associated with these restrictions.
Yes, it will be distributed under a permissive license (e.g., CC BY-SA 4.0) to encourage research use.

**Have any third parties imposed IP-based or other restrictions on the data associated with the instances?**  If so, please describe these restrictions, and provide a link or other access point to, or otherwise reproduce, any relevant licensing terms, as well as any fees associated with these restrictions.
All charts are subjected to their respective copyrights by the authors of this paper.

**Do any export controls or other regulatory restrictions apply to the dataset or to individual instances?**  If so, please describe these restrictions, and provide a link or other access point to, or otherwise reproduce, any supporting documentation.

N/A

**Any other comments?**
Distribution will include detailed usage guidelines to ensure proper application of the dataset.

### H.7. Maintenance

**Who is supporting/hosting/maintaining the dataset?**
The authors of the PDE-Controller framework.

**How can the owner/curator/manager of the dataset be contacted (e.g., email address)?**
Contact information will be provided with the dataset release.

**Is there an erratum?** If so, please provide a link or other access point.
https://pde-controller.github.io/

**Will the dataset be updated (e.g., to correct labeling errors, add new instances, delete instances)?** If so, please describe how often, by whom, and how updates will be communicated to users (e.g., mailing list, GitHub)?
Yes, periodic updates will incorporate additional real-world problems and refinements.

**If the dataset relates to people, are there applicable limits on the retention of the data associated with the instances (e.g., were individuals in question told that their data would be retained for a fixed period of time and then deleted)?** If so, please describe these limits and explain how they will be enforced.
N/A

**Will older versions of the dataset continue to be supported/hosted/maintained?** If so, please describe how. If not, please describe how its obsolescence will be communicated to users.
Yes, previous versions will remain accessible for reproducibility.

**If others want to extend/augment/build on/contribute to the dataset, is there a mechanism for them to do so?** If so, please provide a description. Will these contributions be validated/verified? If so, please describe how. If not, why not? Is there a process for communicating/distributing these contributions to other users? If so, please provide a description.
Yes, contributions will be encouraged through a collaborative platform (e.g., GitHub).

**Any other comments?**
The dataset's maintainers are committed to ensuring its long-term usability and relevance for scientific research.

## I. Misc.

**URL to benchmark.** The benchmark URL can be found here: https://pde-controller.github.io/

**URL to Croissant metadata.** The Croissant metadata URL can be found here: https://huggingface.co/datasets/delta-lab-ai/pde-controller/tree/main

**Author statement & license information.** We the authors bear all responsibility in case of violation of rights.

**Hosting and maintenance.** We have a dedicated webpage for hosting instructions: https://pde-controller.github.io/. We are committed to performing major maintenance periodically.

**Dataset Structure.** All files are stored in the JSONL format. For the translator dataset, we store separate files based on STL syntax formats, the number of constraints, and the train-test split. Each training file contains more than 600 samples, and each test file contains more than 150 samples. Each sample includes the informal question in natural language, the STL

representation in LaTeX, and the corresponding Python script.

For the preference dataset, we split the files based on three difficulty levels and the train-test split. Each sample in the file contains the informal question in natural language, the winner STL that improves the informal question, and the loser STL that worsens it. For each STL, we also provide the resulting utility score.

