# OpenReview forum: "PDE-Controller: LLMs for Autoformalization and Reasoning of PDEs"
_ICML.cc/2025/Conference — ICML 2025 poster_

### Official Review · Reviewer_vDyK · 2025-02-19

**Overall Recommendation:** 4

**Summary:**

This paper presents PDE-Controller, a framework that enables large language models (LLMs) to automate the control of systems governed by partial differential equations (PDEs).

The study highlights the gap between current AI-for-math research, which excels in pure mathematical reasoning, and its limited application in applied mathematics, particularly PDEs. The authors propose a novel pipeline that integrates autoformalization, scientific reasoning, and program synthesis for PDE control. The PDE-Controller translates informal natural language problem descriptions into formal specifications, executes reasoning steps to improve control efficiency, and generates executable code.

The model is trained using supervised fine-tuning (SFT) and reinforcement learning from human feedback (RLHF), leveraging a large dataset of synthetic and human-annotated PDE problems. Experimental results demonstrate that PDE-Controller achieves significant improvements over baseline LLMs in utility gain, autoformalization accuracy, and program synthesis efficiency.

**Claims And Evidence:**

- Results are robust and convincing.

- Claims are supported by the results

**Essential References Not Discussed:**

Might worth reviewing and maybe mentioning:
Explain Like I'm Five: Using LLMs to Improve PDE Surrogate Models with Text (arXiv preprint arXiv:2410.01137)
author: Lorsung, Cooper and Farimani, Amir Barati

**Experimental Designs Or Analyses:**

- The experimental design relies heavily on synthetic data generation from a limited set of template rules. This may pose challenges when generalizing to different formats.

- The comparison to established (non-learning) methods is lacking. It is unclear how much this approach improves beyond existing human-centric methods, whether in terms of labor reduction or accuracy.

**Methods And Evaluation Criteria:**

Methods and evaluation criteria are clearly defined. They seem to be comprehensive and complete.

**Other Comments Or Suggestions:**

- Line 216 (Left) – pairs -> triplets

- Line 317 (Left) – and and

- Figure 6 – the meaning of A, B and C (i.e., the constraints?) are not explained

**Other Strengths And Weaknesses:**

- **Strengths:**
  i. The authors explicitly address LLM reasoning by introducing sub-goal generation and optimization, which were not originally present, and compare the results achieved with and without these components.
  ii. The study leverages both SFT and RLHF to improve performance when utilizing sub-goals.
  iii. The paper is well-written, with a clear and concise presentation that is easy to read and follow.

- **Weaknesses:**
  i. *(Lines 430-431)* Failures on real manual data highlight the proposed method's limitations in generalizing to real-world cases and its reliance on synthetic data generated from a limited set of templates. However, the fact that other models also fail suggests that the proposed method has merits, even if its capabilities remain constrained.

**Questions For Authors:**

No further questions.

**Relation To Broader Scientific Literature:**

The work utilizes state-of-the-art methods and models, demonstrating clear advantages when applying these advanced techniques to new domains.

**Theoretical Claims:**

There are no clear theoretical claims. The work is mainly focused on adapting known methods and approaches to a new, unexplored domain.

---

> ### Author Rebuttal · Authors · 2025-04-01
>
> We deeply appreciate your feedback and suggestions.
>
> > 1. The experimental design relies heavily on synthetic data generation from a limited set of template rules. This may pose challenges when generalizing to different formats.
>
> We fine-tune our models on synthetic data mainly because it is time and resource consuming to manually curate data. It took 17 human-hours to collect 17 heat and 17 wave problems. We will try our best to collect more manually written samples in the next step.
> It is possible to scale our method to more than three STLs; there are no additional technical challenges. We limited our dataset to three constraints to balance between exhaustive coverage of the logic formulations with computation constraints: in total we spent about two months on preparing our dataset, including collecting 2 million samples, data augmentations, and PDE simulations.
>
> Due to limited time during the rebuttal, below we show the test performance on generalizing to unseen 4-constraint STLs of our Translator (IoU) and Coder (executability & utility), over 5 problems each for heat and wave problems. Note that both models have never seen problems with 4 STLs. MathCoder2 is evaluated with 2-shot in-context examples of 4 constraints giving it an advantage. We will include these results in our camera-ready.
>
> |PDE|Model|IoU (Translator)|Executability($\uparrow$) (Coder)|UtilityRMSE($\downarrow$)|
> |----------|----------|----------|----------|----------|
> |Heat|Ours|0.934 (0.0)|0.8 (0.0)|0.0 (0.0)|
> |Heat|MathCoder2|0.8154 (0.0)|0.6 (0.0)|0.2600 (0.0)|
> |Wave|Ours|1.0 (0.0)|0.8 (0.0)|0.1515 (0.0)|
> |Wave|MathCoder2|0.9690 (0.0)|0.8 (0.0)|0.2393 (0.2268)|
>
> > 2. The comparison to established (non-learning) methods is lacking. It is unclear how much this approach improves beyond existing human-centric methods, whether in terms of labor reduction or accuracy.
>
> Due to the time and resource demands of having individual human experts manually formulate a given problem, code and optimizable problem and potentially reason about subgoals, collecting a large number of human-centric solutions is difficult. We will try our best to include this comparison in the camera-ready version.
>
>
> > 3. Might worth reviewing and maybe mentioning: Explain Like I'm Five: Using LLMs to Improve PDE Surrogate Models with Text (arXiv preprint arXiv:2410.01137) author: Lorsung, Cooper and Farimani, Amir Barati
>
> Thank you for your suggestion. We will add this to our related work.
>
> > Other Comments Or Suggestions:
> > * Line 216 (Left) – pairs -> triplets
> > * Line 317 (Left) – and and
> > * Figure 6 – the meaning of A, B and C (i.e., the constraints?) are not explained
>
> Thank you for your comments. We will correct and clarify these for the camera ready.

---

### Official Review · Reviewer_bygn · 2025-03-10

**Overall Recommendation:** 4

**Summary:**

The paper introduces PDE-Controller, a framework leveraging large language models (LLMs) for automating the formalization and reasoning of control problems governed by partial differential equations (PDEs). The authors claim significant performance improvements in translating informal natural language PDE control problems into formal specifications using Signal Temporal Logic (STL), synthesizing executable Python code, and proposing effective intermediate reasoning subgoals. Experimental results demonstrate up to 62% improvement in PDE control utility over baseline LLM models, supported by a newly created dataset comprising over 2 million samples.

**Claims And Evidence:**

- Claim: PDE-Controller significantly outperforms baseline models in PDE control reasoning.
  - Evidence: Demonstrated 62% improvement in utility gain compared to GPT-4o and MathCoder2.

- Claim: PDE-Controller effectively formalizes informal PDE problems into STL and Python code.
  - Evidence: Achieves autoformalization accuracy of over 64% and program synthesis accuracy over 82%.

- Claim: The Controller model effectively decomposes complex PDE problems into manageable subgoals.
  - Evidence: Empirical results show higher success rates and substantial improvements in utility using subgoal decomposition compared to random and baseline models.

**Essential References Not Discussed:**

The paper sufficiently addresses related works but could further discuss recent developments in differentiable physics and physics-informed neural networks (PINNs) which also address PDE control.

**Experimental Designs Or Analyses:**

Experiments were conducted on 1D heat and wave PDE problems. The authors performed comprehensive evaluations using various metrics (IoU, executability, utility RMSE), comparing PDE-Controller against established baselines (MathCoder2, GPT-4o). The designs are sound, adequately controlled, and effectively validate the proposed model’s strengths.

**Methods And Evaluation Criteria:**

The proposed methods and evaluation criteria, including autoformalization, subgoal reasoning via reinforcement learning from human feedback, and synthesis of executable Python programs, are appropriately tailored for PDE control tasks. The benchmarks and metrics (IoU, executability, utility RMSE) are appropriate and effectively capture the nuances of PDE control scenarios.

**Other Comments Or Suggestions:**

Consider additional comparisons with differentiable physics or physics-informed neural networks for completeness.

**Other Strengths And Weaknesses:**

Strengths:

- Innovative use of LLMs for formalization and reasoning in PDE control.

Weaknesses:

- Dependence on external optimization solvers (e.g., Gurobi) limits standalone applicability.

- Limited exploration beyond 1D problems.

**Questions For Authors:**

- How would your method scale to multi-dimensional PDE problems?

- Have you evaluated or planned internal optimization methods to reduce dependency on external solvers like Gurobi?

- How does your approach handle poorly formulated or noisy natural language inputs in real-world scenarios?

**Relation To Broader Scientific Literature:**

The contributions relate closely to recent advances in AI-for-math and PDE control literature, particularly highlighting the gap between general-purpose LLMs and specialized scientific reasoning capabilities. The use of STL for formalization and the combination of reinforcement learning with human feedback align well with contemporary approaches in both AI-for-science and formal methods.

**Theoretical Claims:**

The paper does not explicitly focus on new theoretical proofs but instead emphasizes methodological innovations and empirical validations. Hence, there are no direct theoretical proofs to evaluate.

---

> ### Author Rebuttal · Authors · 2025-04-01
>
> We deeply appreciate your feedback and suggestions.
>
> > The paper sufficiently addresses related works but could further discuss recent developments in differentiable physics and physics-informed neural networks (PINNs) which also address PDE control.
>
> Thank you for the suggestion! We will include a discussion on recent developments in differentiable physics and physics-informed neural networks (PINNs) in the related work section [1, 2]. It will be an interesting study to replace our numerical solver with neural operators in our framework.
>
> [1] “Learning to control pdes with differentiable physics”
>
> [2] “Solving pde-constrained control problems using operator learning”
>
>
> > Q1. How would your method scale to multi-dimensional PDE problems?
>
> Extending our method to multi-dimensional PDEs, such as 2D Navier Stokes, will involve enhancing the Gurobi solver, which is a topic for our future work. Nonetheless, the development of more advanced solvers will not impact our core contributions to the design and fine-tuning of our LLMs.
>
>
> > Q2. Have you evaluated or planned internal optimization methods to reduce dependency on external solvers like Gurobi?
>
> We indeed plan to develop internal optimizers for more general PDE settings, to reduce our dependence on external solvers.
> Meanwhile, we may not completely eliminate the use of external solvers. Integrating well-developed external solvers into LLMs can be a strength in solving complex problems, as demonstrated in the following examples:
> 1) In theorem proving, any LLMs depend on their proof environments to construct proofs. [1] integrates their LLM based prover with the Lean proof environment to present promising premises for interactive proof generation. [2] enables LLMs that integrate with Lean for tactic suggestion, proof search, and premise selection. [3] and [4] are further works that require Lean.
> 2) Moreover [5] presents an LLM framework that autoformalizes natural language linear programming problems then calls an external optimizer for optimization.
>
>
> [1] Yang, K., Swope, A. M., Gu, A., Chalamala, R., Song, P., Yu, S., Godil, S., Prenger, R., & Anandkumar, A. (2023). LeanDojo: Theorem Proving with Retrieval-Augmented Language Models. arXiv preprint arXiv:2306.15626.
>
> [2] Song, P., Yang, K., & Anandkumar, A. (2025). Lean Copilot: Large Language Models as Copilots for Theorem Proving in Lean. arXiv preprint arXiv:2404.12534
>
> [3] Wang, R., Zhang, J., Jia, Y., Pan, R., Diao, S., Pi, R., & Zhang, T. (2024). TheoremLlama: Transforming General-Purpose LLMs into Lean4 Experts. arXiv preprint arXiv:2407.03203.
>
> [4] Lin, Y., Tang, S., Lyu, B., Wu, J., Lin, H., Yang, K., Li, J., Xia, M., Chen, D., Arora, S., & Jin, C. (2025). Goedel-Prover: A Frontier Model for Open-Source Automated Theorem Proving. arXiv preprint arXiv:2502.07640.
>
> [5]  Zhang, J., Wang, W., Guo, S., Wang, L., Lin, F., Yang, C., & Yin, W. (2024). Solving General Natural-Language-Description Optimization Problems with Large Language Models. arXiv preprint arXiv:2407.07924
>
>
>
>
> > Q3. How does your approach handle poorly formulated or noisy natural language inputs in real-world scenarios?
>
> Currently, we add robustness in our LLM against noises in real-world scenarios via natural language data augmentation with chatGPT. To improve our future research, we plan to introduce more structure-level augmentations to our synthetic data for additional robustness against noisy inputs. We fine-tune our models on synthetic data mainly because it is time and resource consuming to manually curate data. It took 17 human-hours to collect 17 heat and 17 wave problems.

---

### Official Review · Reviewer_pp3p · 2025-03-12

**Overall Recommendation:** 5

**Summary:**

The paper proposes that the PDE-Controller framework enhances large language models (LLMs) to control systems governed by PDEs, addressing their limitations in rigorous logical reasoning. It transforms natural language instructions into formal specifications, improving PDE control's reasoning, planning, and utility. The holistic solution includes datasets, math-reasoning models, and novel metrics, outperforming existing models by up to 62\% in utility gain. This work bridges language generation and PDE systems, showcasing LLMs' potential in scientific and engineering applications.

**Claims And Evidence:**

The claims presented in the paper are well-supported by empirical evidence.

**Essential References Not Discussed:**

The paper adequately discusses relevant prior work.

**Experimental Designs Or Analyses:**

The experimental design is well-justified, supporting the claims regarding the effectiveness of LLMs in PDE control. The authors thoroughly analyze the framework's impact on reasoning and control performance.

**Methods And Evaluation Criteria:**

The proposed methodology and evaluation criteria are well-structured and appropriate for assessing the framework's performance.

**Other Comments Or Suggestions:**

N/A

**Other Strengths And Weaknesses:**

1) Novel framework for PDE control automation: The paper introduces an innovative approach that leverages LLMs for reasoning-based PDE control.

2) New dataset: The dataset enables evaluating LLMs' reasoning capabilities in PDE control scenarios.

3) Significant performance improvements: The framework outperforms existing models considerably.

**Questions For Authors:**

1) Have you explored alternative reinforcement learning algorithms besides DPO for training the controller? Given recent advancements in reasoning-enhanced LLMs, comparing the performance of GRPO or other RL-based methods in training the PDE controller would be valuable.

**Relation To Broader Scientific Literature:**

This research presents a novel contribution at the intersection of LLMs and applied mathematics, particularly in PDE control. This area has received limited attention in the literature.

**Theoretical Claims:**

No explicit theoretical claims are presented in the paper.

---

> ### Author Rebuttal · Authors · 2025-04-01
>
> We deeply appreciate your feedback and suggestions.
>
> > Have you explored alternative reinforcement learning algorithms besides DPO for training the controller? Given recent advancements in reasoning-enhanced LLMs, comparing the performance of GRPO or other RL-based methods in training the PDE controller would be valuable.
>
> For alternative RL algorithms besides DPO, we also experimented with Eq. 3 without the SFT regularization term and found that it led to degraded generation and overfitting. Our Eq. 3 is stable and achieves strong performance. We will include this discussion in the camera ready.
> We agree that further study of GRPO [1] and the latest RL methods is valuable, and we plan to explore them in our Controller training in future work.
>
> [1] “DeepSeek-R1: Incentivizing Reasoning Capability in LLMs via Reinforcement Learning” DeepSeek-AI (Arxiv 2501.12948)

---

> > ### Comment · Reviewer_pp3p · 2025-04-02
> >
> > I thank the author for the rebuttal. After going through the rebuttal response and other reviewer responses, I would like to increase my score with the hope that the author will incorporate all the rebuttal responses in the final version of the manuscript.

---

### Official Review · Reviewer_kzDp · 2025-03-22

**Overall Recommendation:** 1

**Summary:**

This paper develops PDE-Controller that uses LLMs to solve open-loop control inputs for PDEs with constraints. The PDE-Controller uses LLMs to transform informal natural language instructions into formal specifications in the form of STL, and then combine optimization solvers with LLM reasoning to improve the utility of PDE control. It has been observed that the PDE-Controller significantly outperforms GPT4o and a few open-source models in utility gain for PDE control.

**Claims And Evidence:**

To demonstrate the advantage of PDE-Controller, the authors got 2 million synthetic samples for the control of heat and wave equations, and gathered another 34 human-written problems. Then the authors developed various metrics such as IoU and Utility RMSE and use such metrics to support their claim.

Based on their results, I am convinced that their proposed PDE-Controller framework can achieve good utility.

**Essential References Not Discussed:**

There is a large body of textbooks and papers on closed-loop PDE control, which are not mentioned by this paper. If the authors do not want to touch on closed-loop PDE control, maybe it is worth revising the paper to emphasize this at the beginning of the paper. I mean, from what I understand, when control people talk about "PDE control", they typically mean "closed-loop PDE control."

**Experimental Designs Or Analyses:**

I read all the experimental results. The metrics make sense, and the results are solid. However, I have also commented on a few things that I think are missing.

**Methods And Evaluation Criteria:**

The metrics proposed in this paper make sense to me. There may be several other metrics that are needed. For example, it may be beneficial if the authors can include another metric that measures whether the STL are fully met by the solutions. In addition, there is a gap between the metrics in this paper and the standard pass@k metrics used in the LLM literature.

On evaluation methods, some ablation study is also missing. There are a few components for PDE-Controller. It is unclear whether all these components are that essential, and some ablation study could be helpful.

Finally, it may be useful if the authors can include one reasoning model (e.g. o1) for their evaluations.

**Other Comments Or Suggestions:**

The paper is well written and I have not noticed many typos.

**Other Strengths And Weaknesses:**

The way how this paper uses LLMs for PDE control is original. The significance is questionable since the paper does not consider closed-loop control, more complicated PDEs, as well as the case with more than three STLs.

**Questions For Authors:**

1. It may be beneficial if the authors can include another metric that measures whether the STL are fully met by the solutions. In addition, is there a way to connect the evaluations with the standard pass@k metrics used in the LLM literature?

2. Have the authors considered doing a comprehensive ablation study?

3. Have the authors considered adding a one reasoning model (e.g. o1) as a strong baseline?

4. Closed-loop PDE control is typically preferred for real systems. Have the authors considered closed-loop control?

5. How are the utility and STLs used in the formulation of this paper connected to the traditional PDE control objectives such as setpoint/trajectory tracking and disturbance rejection?

6. Can the authors extend their method for more complicated PDEs such as 2D NS equations?

7. Can the authors comment on the possibility of scaling their method for more than three STLs?

**Relation To Broader Scientific Literature:**

Overall, this paper is relevant to the big area of LLMs for science and engineering. However, the scope of this paper is confined to a very specific question: how to generate open-loop control for heat or wave equations with up to three STLs. The scope is quite narrow. Clearly, the paper will be significantly improved if they can i) consider closed-loop control, ii) consider more complicated PDEs (e.g, 2D NS equation), iii) scale for more than three STLs. I am mostly concerned regarding the first item. Due to the uncertainty of real systems, sensing, actuation, and feedback are typically needed to deploy closed-loop PDE control. This paper completely ignores this issue. It is also unclear to me how the utility and STLs used in the formulation of this paper are connected to the traditional PDE control objectives such as setpoint/trajectory tracking and disturbance rejection.

**Theoretical Claims:**

This paper does not have theoretical contributions.

---

> ### Author Rebuttal · Authors · 2025-04-01
>
> We deeply appreciate your feedback and suggestions.
> > Q1
> 1) Our utility score (A.2) can faithfully quantify whether STL (constraints) are fully met by solutions simulated by the solver. This utility is inherited from [1] and serves as the rule-of-thumb “accuracy” metric. pp3p affirms “evaluation criteria are well-structured and appropriate for assessing the framework's performance” and Bygn: “metrics [...] are appropriate and effectively capture the nuances of PDE control scenarios”.
> 2) We further explain the connection b/w our metrics and pass@k below:
> * For autoformalization (Translator), IoU is equivalent to “average@k”, it averages the alignment b/w predictions and targets over multiple generations and problems. We can discretize IoU into “pass@k” by considering only pass/fail cases based on token-level differences; but this ignores fine-grained quantifications of autoformalization. In the tables below, IoU and pass@k are not always aligned.
> * For code generation (Coder), the executability metric is essentially pass@k from the executability perspective.
>
> Heat
> |Model|IoU|Pass@1|pass@2|pass@3|
> |---|---|---|---|---|
> |Ours|0.992(0.07)|0.978(0.142)|0.980(0.134)|0.982(0.131)|
> |MathCoder2|0.772(0.35)|0.538(0.480)|0.565(0.484)|0.583(0.493)|
>
> Wave
> |Model|IoU|Pass@1|pass@2|pass@3|
> |---|---|---|---|---|
> |Ours|0.992(0.07)|0.971(0.161)|0.975(0.152)|0.977(0.149)|
> |MathCoder2|0.1953(0.045)|0.3305(0.4396)|0.3726(0.4663)|0.3971(0.4893)||
>
> [1] "Formal Methods for Partial Differential Equations" Alvarez 2020.
> > Q2
>
> Yes, vDyK affirms: “Methods and evaluation criteria are clearly defined”. Bygn: “The designs are sound, adequately controlled, and effectively validate the proposed model’s strengths.” To demonstrate that our components are essential, we clarify the ablation comparisons below based on Heat problem results (Table 3 & 10):
> * Our Translator component is essential. It has better autoformalization abilities; +28.5% IoU over MathCoder2.
> * Our Coder component is essential and robust to noisy autoformalization:
>   * Given ground truth STL, our Coder has +4% better executability rate in generated Python than MathCoder2, and utility RMSE is 91.6% lower (better) than MathCoder2.
>   * When switching from noisy Translator predictions to ground truth STL, our Coder’s utility only drops 0.57% indicating that our Coder is robust under noisy STL inputs. We will include this discussion in our camera ready.
> |Method|Performance|
> |---|---|
> |Mathcoder’s translator(Table3)|IoU=0.772|
> |Our Translator(Table3)|IoU=0.992(+28.5%)|
> |Mathcoder’s coder(Table3)|Executability=0.9592|
> |Our Coder(Table3)|Executability=0.9978(+4.02%)|
> |Mathcoder’s coder(Table3)|UtilityRMSE=0.2058|
> |Our Coder(Table3)|UtilityRMSE=0.0173(-91.6%)|
> |TranslatorSTL→Coder(Table10)|UtilityRMSE=0.0174|
> |ground truth STL→Coder(Table3)|UtilityRMSE=0.0173(-0.57%)|
>
> > Q3
>
> We have added the o1-mini reasoning model (Table 3, 4, 6). This cost-efficient version, suitable for our academic budget, performs comparably to the o1 model in math and coding tasks. OpenAI’s official statement: “o1-mini may outperform o1-preview when it comes to coding applications” [1] [2]. Further “o1‑mini excels at STEM, especially math and coding” and “for applications that require reasoning without broad world knowledge” [3].
>
> [1] [OpenAI](https://help.openai.com/en/articles/9855712-openai-o1-models-faq-chatgpt-enterprise-and-edu)
>
> [2] [Benchmark test](https://aimlapi.com/comparisons/openais-o1-preview-vs-o1-mini)
>
> [3] [OpenAI](https://openai.com/index/openai-o1-mini-advancing-cost-efficient-reasoning/)
>
> [4] and Table 1 and Fig. 6 in Wu, S., et al. 2024. A Comparative Study on Reasoning Patterns of OpenAI's o1 Model. arXiv:2410.13639
> > Q4
>
> Thank you for raising this important issue. Indeed, closed-loop control is more realistic in applications. It is among future routes we plan to explore. Extending to closed-loop control is possible by appending the utility from the subgoal optimization into future optimization rounds. But this increases the complexity of LLM fine-tuning. Thus, as the first step in this direction we focus on open-loop control. We will include this discussion and provide clarification in our camera ready.
> > Q5
>
> 1) Both setpoint tracking and our solver (A.2~A.3) require discretizing PDEs and constraints and designing cost functions or utility scores to characterize PDE control objective satisfaction. The core differences lie in the cost functions’ or utility scores’ design and formulation. Our utility score can better handle inequality constraints (A.2) compared to tracking errors, which mainly aim to reduce distance to the target.
> 2) Our work does not explicitly model disturbances such as (thermal) noise or variations in material properties (e.g., diffusivity).
> Overall, we see no key barriers to replacing our current solver with those for setpoint tracking or disturbance rejection.
>
> > Q6
>
> Please see Bygn Q1 (due to character limits).
> > Q7
>
> Please see vDyK Q1 (due to character limits).

---

### Decision · Program_Chairs · 2025-05-01

**Decision:**

Accept (poster)

**Comment:**

This paper introduces "PDE-Controller," a novel framework leveraging Large Language Models (LLMs) for the challenging task of formalizing, reasoning about, and controlling Partial Differential Equations (PDEs)-governed systems. The authors propose a pipeline involving natural language understanding, translation to formal specifications (STL), subgoal reasoning, and code generation, supported by new datasets and evaluation metrics. The work demonstrates significant performance improvements over strong baselines like GPT-4o on heat and wave equation control problems.

The paper received generally positive reviews, with three out of four reviewers recommending acceptance. Reviewers broadly agreed on the paper's strengths, including the novelty of applying LLMs to PDE control automation (pp3p, bygn, vDyK), the creation of valuable datasets (human and large-scale synthetic) (pp3p), the significant empirical performance improvements shown (pp3p, bygn), and the well-structured methodology and evaluation (pp3p, bygn, vDyK). The work is seen as a valuable contribution at the intersection of LLMs and applied mathematics/scientific computing.

Reviewer kzDp raised concerns regarding the scope (limited to open-loop control, specific 1D PDEs, few constraints), the connection to traditional control objectives, and the need for specific metrics (pass@k, STL satisfaction) and ablation studies. The authors provided a detailed and rebuttal addressing these concerns. They clarified how their utility metric relates to STL satisfaction and pass@k, pointed to existing results for ablation evidence, added a requested baseline (o1-mini), justified the open-loop focus as a first step towards closed-loop control, discussed scalability (including preliminary positive results on 4-constraint problems), and addressed the role of external solvers and data limitations. It looks like the concerns are addressed, despite that reviewer didn't have a chance to update the review post-rebuttal.

Based on the majority support from the reviewers, the convincing and detailed author rebuttal that addressed the raised concerns, i recommend accepting this paper.